# Limit state equation and failure pressure prediction model of pipeline with complex loading

Ming-ming Sun [1,2,3], Hong-yuan Fang[1,2,3] ✉, Nian-nian Wang[1,2,3], Xue-ming Du[1,2,3], Hai-sheng Zhao[4,5] & Ke-Jie Zhai[1,2,3]

Assessing failure pressure is critical in determining pipeline integrity. Current research primarily concerns the buckling performance of pressurized pipelines subjected to a bending load or axial compression force, with some also looking at the failure pressure of corroded pipelines. However, there is currently a lack of limit state models for pressurized pipelines with bending moments and axial forces. In this study, based on the unified yield criterion, we propose a limit state equation for steel pipes under various loads. The most common operating loads on buried pipelines are bending moment, internal pressure, and axial force. The proposed limit state equation for intact pipelines is based on a three-dimensional pipeline stress model with complex load coupling. Using failure data, we investigate the applicability of various yield criteria in assessing the failure pressure of pipelines with complex loads. We show that the evaluation model can be effectively used as a theoretical solution for assessing the failure pressure in such circumstances and for selecting appropriate yield criteria based on load condition differences.

Due to their large capacity, high bearing capacity, and low environmental impact, steel pipelines are widely used as a safe and cost-effective material for long-distance water diversion, urban heating, and transportation of oil, gas, and other materials[1–3]. Internal pressure determines the size of the pipeline and the efficiency with which material is transported[4–6]. Currently, the largest diameter of a pressurized pipe is 1200 mm, and as pipeline transportation distance increases, so does the operating pressure, which can reach more than 10 MPa. For crude oil and other mediums with high viscosity and freezing points, the transportation pressure can reach 20 MPa to ensure normal transportation. We can infer that assessing failure pressure is critical in determining pipeline integrity.

During operation, buried pipelines are subjected to combined loads such as axial force, internal pressure, and bending moment[7–9].

Internal pressure in buried pipelines is primarily caused by the internal transport medium, with axial force resulting from temperature differences during laying and operation. Geotechnical disturbances may cause axial forces within the pipeline. For example, when pipelines cross a slope, the pipelines at the bottom withstand axial compression while those at the top withstand axial tension[10]. The Poisson effect is another type of axial force in materials. Because of the pressure inside the pipeline, radial expansion will attempt to shorten the pipeline's axial length. Pipelines may also experience bending moments due to soil movement caused by landslides, settlements, frost heave, earthquakes, foundation subsidence, and debris flows.

In this case, mechanical model analysis and bearing capacity evaluation of intact pipelines are the foundation for evaluating pipeline system integrity[11]. Taylor et al.[12] demonstrated that an additional

[1]School of Water Conservancy and Transportation, Zhengzhou University, Zhengzhou 450001, China. [2]National Local Joint Engineering Laboratory of Major Infrastructure Testing and Rehabilitation Technology, Zhengzhou 450001, China. [3]Collaborative Innovation Center of Water Conservancy and Transportation Infrastructure Safety, Zhengzhou 450001, China. [4]State Key Laboratory of Coastal and Offshore Engineering, Dalian University of Technology, Dalian 116024, China. [5]School of Hydraulic Engineering, Faculty of Infrastructure Engineering, Dalian University of Technology, Dalian 116024, China. ✉e-mail: fanghongyuan1982@163.com

bending load reduced the failure pressure. Similar conclusions could be drawn for pipelines that could withstand axial forces[13,14]. Therefore, assessing failure pressure solely based on internal pressure is limited, and it is necessary to examine the burst pressure of pipelines carrying complex loads.

According to reports, there is little literature on the integrity assessment of pipelines carrying complex loads. Early research focused on determining the bearing capacity of pipelines under two types of loads: axial force and internal pressure. Some studies[15–22] referred to the research project of the Southwest Research Institute (SwRI) from the 1990s. Other references[23–28] discussed the "reliability of corroded pipes" in the Joint Industry Projects (JIP) developed by DET NORSKE VERITAS (DNV). DNV's method modifies the failure pressure model by including the stress factor $H_1$. Without safety factors, the failure pressure $p_f$ of an intact pipeline is calculated as follows:

$$p_f = \frac{2t\sigma_u}{D_e - t} H_1 \tag{1}$$

$$H_1 = 2\left(1 + \frac{\sigma_L}{\sigma_u}\right) \tag{2}$$

Based on the above formulae, $D_e$ is the diameter, $t$ is the wall thickness, $\sigma_L$ is the axial stress, and $\sigma_u$ is the tensile strength. The above model is appropriate as corrosion defects can withstand compressive stress. However, increasing longitudinal stresses in the corrosion region ($\sigma_L < 0.25\sigma_u$) significantly decreases prediction accuracy[29].

Zhao et al.[30] proposed a prediction model for determining the burst pressure of a pipeline under axial load. However, this model assumes that the pipe section is subjected to the same axial stress, so it does not apply to the uneven axial stress caused by the bending moment. Zhou et al.[31] discovered that axial stress reduced failure pressure and developed a vertical strain-based prediction formula. However, it is only applied to intact pipelines or pipelines with minor defects. Chen et al.[32] first proposed a semi-empirical formula for calculating the failure pressure of pipelines subjected to internal pressure and axial tension, improving the theory of evaluating pipelines' bearing capacity under complex loads. However, this method does not consider bending moment loads, and there is a need to confirm its accuracy for pipelines with other defect types.

Researchers have proposed more complex failure pressure assessment schemes to address the shortcomings of existing evaluation models. Chauhan and Swankee[33] and Liu et al.[34] proposed interactive charts to estimate the residual strength of pipelines under external loads. However, this method does not apply to pipelines with bending moments and axial forces. Heitzer[35] presented a mathematical programming formulation and used a numerical procedure to analyze pipeline plastic collapse under internal pressure and axial force. Benjamin[36] and Bruère et al.[37] proposed an approach to assess failure pressure that considers the impact of axial compressive force. Arumugam et al.[38] utilized the FE method to determine the bearing capacity of colony corrosion defects under compressive load and internal pressure. Zhang and Zhou[39] proposed an artificial neural network-based model for assessing the internal pressure bearing capability of pipelines subjected to axial force. Shuai et al.[40] applied finite element analysis to investigate the buckling bearing capacity of pipelines under axial compressive loading and internal pressure. Konosu and Mukaimachi[41] put forward a plastic collapse assessment procedure for pressurized vessels with bending moments. Mohd et al.[42] investigated the same pipeline case.

The FE method and failure tests determine the residual bearing capacity of pressurized pipelines with bending or multiple external loads. Mondal and Dhar[29] applied the finite element method to model marine pipelines and assess their structural integrity. Chegeni et al.[43] analyzed the effect of corrosion damage on the bearing capacity of

**Table 1 | Calculation equation of failure pressure**

| case | additional load | principal stress | judging condition | Calculation equation |
|------|-----------------|------------------|-------------------|----------------------|
| Case 1 | positive | $\sigma_1 = \sigma_L,$ $\sigma_2 = \sigma_h,$ $\sigma_3 = 0.$ | $\sigma_2 \le \frac{\sigma_1 + \sigma_3}{2}$ | $\left(\frac{1}{2} - \frac{b}{1+b}\right)\frac{p}{p_0} + \frac{\sum(\sigma_L)}{\sigma_u} = 1$ |
| Case 2 | | | $\sigma_2 > \frac{\sigma_1 + \sigma_3}{2}$ | $\left(\frac{1}{2(1+b)} + \frac{b}{b+1}\right)\frac{p}{p_0} + \frac{1}{1+b}\frac{\sum(\sigma_L)}{\sigma_u} = 1$ |
| Case 3 | | $\sigma_1 = \sigma_h,$ $\sigma_2 = \sigma_L,$ $\sigma_3 = 0$ | - | $\frac{1}{1+b}\left(1 + \frac{b}{2}\right)\frac{p}{p_0} + \frac{b}{1+b}\frac{\sum(\sigma_L)}{\sigma_u} = 1$ |
| Case 4 | negative | $\sigma_1 = \sigma_h,$ $\sigma_2 = 0,$ $\sigma_3 = \sigma_L$ | $\sigma_2 \le \frac{\sigma_1 + \sigma_3}{2}$ | $\left(1 - \frac{1}{2(1+b)}\right)\frac{p}{p_0} + \frac{1}{1+b}\frac{\sum(\sigma_L)}{\sigma_u} = 1$ |
| Case 5 | | | $\sigma_2 > \frac{\sigma_1 + \sigma_3}{2}$ | $\left(\frac{1}{b+1} - \frac{1}{2}\right)\frac{p}{p_0} + \frac{\sum(\sigma_L)}{\sigma_u} = 1$ |
| Case 6 | | $\sigma_1 = \sigma_h,$ $\sigma_2 = \sigma_L,$ $\sigma_3 = 0$ | - | $\left(1 - \frac{b}{2(1+b)}\right)\frac{p}{p_0} + \frac{b}{1+b}\frac{\sum(\sigma_L)}{\sigma_u} = 1$ |

$\sigma_1$, $\sigma_2$, and $\sigma_3$ represent three types of principal stresses; $\sigma_h$ and $\sigma_L$ respectively denote circumferential and axial stresses; $\sigma_u$ represents the tensile strength; $p$ is the internal pressure load; $p_0$ denotes the ultimate bearing capacity of the internal pressure (only the internal pressure load is applied); and $b$ is the yield criterion parameter.

pressurized pipelines under bending moment loads. Gao et al.[44] investigated the bending bearing capacity of pipelines carrying multiple loads. Similar studies were also conducted by ref. [29,45,46] Luigino et al.[47] described the key findings of the HOTPIPE project, which aimed to determine the bending moment capacity. Ozkan and Mohareb[48,49] examined moment resistance through finite element modeling and full-scale experiments. Smith and Grigory[17] proposed global buckling failure envelopes for pipelines with combined stresses. Roy et al.[22]. put forward a theoretical method to assess the integrity of pipelines with multiple loads. Agarwal[50] conducted a parametric study of pipelines with bending moment and pull force and proposed an optimization method for thickness distribution around them.

According to the literature review, the current research primarily concerns the buckling performance of pressurized pipelines subjected to a bending load or axial compression force. Some studies looked at the failure pressure of corroded pipelines using individual bending moments or axial compressive forces. However, there is currently no limit state model for pressurized pipelines with bending moments and axial forces.

This study presents the limit state equation of pipelines with complex loads based on the unified yield criterion (UYC) theory. This equation can calculate the failure pressure of intact pipelines under multiple or individual external loads (e.g., bending moment, axial compressive force, axial tensile force, and internal pressure). Moreover, this study combines the full-scale pressurized pipeline burst test with axial force to clarify the pipeline's failure mechanism with complex loads. As combined with the burst test data, the accuracy of the limit state equation is verified.

## Results
### The limit state equation of pipeline with complex loading
Table 1 summarizes the limit state equations of pipelines under various conditions that can be utilized to solve the burst pressure with different yield criteria.

### Analysis of experimental results
Figure 1 depicts photographs of intact pipeline explosion failures. The middle lower part of the pipeline burst, immediately tearing adjacent pipe walls, forming a burst opening, and causing noticeable bulging at the failure site.

As shown in Fig. 2a, the wall thickness distribution varied significantly in the circumferential direction while fluctuating minimally in the axial. The lower part of the pipeline in the circumferential direction of 252–324° had the smallest wall thickness, with an average

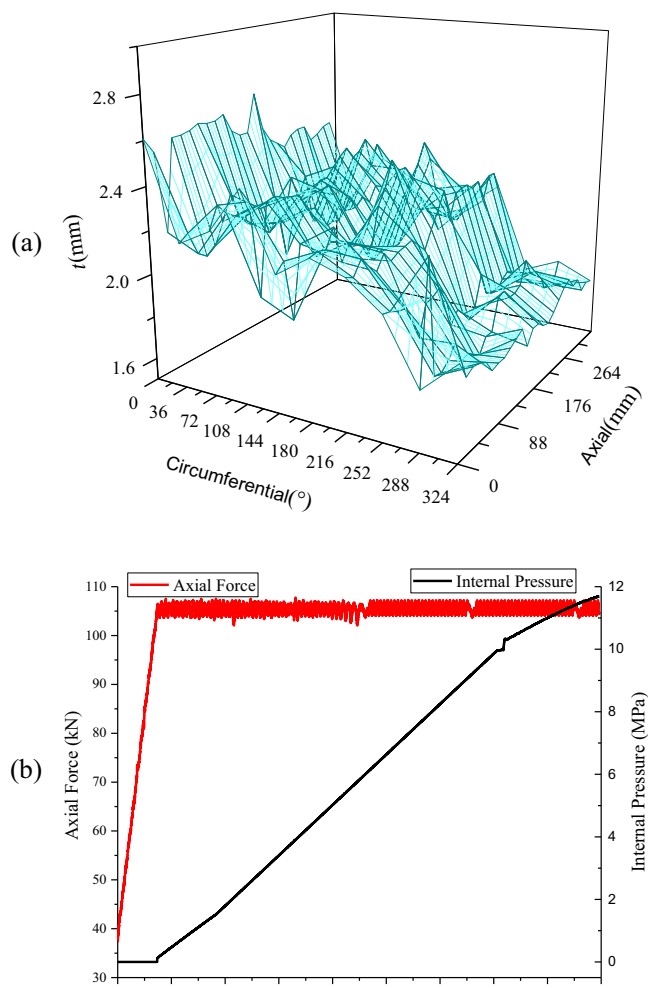

**Fig. 1 | Failure modes of intact pipelines. a, b** Global and partial failure diagrams of the pipeline test, respectively. The location of the pipeline burst failure exhibited significant expansion and shear failure under internal pressure.

**Fig. 2 | Test pipeline data. a** Geometry distribution of wall thickness. The figure shows the difference between the circumferential and axial distributions of pipeline wall thickness. The pipeline's average wall thickness was 2.113 mm. Wall thickness had an average variance of 0.095 in the axial direction and 0.238 in the circumferential direction. In this case, the difference in wall thickness in the circumferential direction was more significant. In the circumferential direction, the wall thickness values within the 324° to 36° range were relatively large (average 2.49 mm). In contrast, the wall thickness values within the 252° to 324° range were relatively small (average of 1.81 mm). **b** Test loading curve. The load curve of the test depicts the various types and values of loads the pipeline could withstand at any loading time. Within 0–147 s, the internal pressure (0 MPa) remained constant while the axial load gradually increased from 37.5 kN to 105.5 kN. From 147 s to 1800 s, the axial load (105.5 kN) remained constant, while internal pressure increased until the pipeline failed. The pipeline's failure loads included an axial tension force of 68 kN and an internal pressure of 11.6 MPa.

of 1.81 mm, making it the most vulnerable to damage. The actual blasting position was in the middle and lower half of the 270° circumferential direction, which confirmed the wall thickness measurement results.

Figure 2b depicts the loading curve of an intact pipeline. Given that the loading device's unloaded gravity was 37.5 kN, it was initially loaded to 37.5 kN to eliminate its impact. Then, the axial force was increased to 37.5 + 68 = 105.5 kN to simulate the increase in axial force caused by temperature differences. After being loaded to 105.5 kN, the axial force remained constant, and internal pressure was applied. The failure pressure for a pipeline explosion was 11.6 MPa.

Figure 3 presents the strain curve with axial force and internal pressure, while Fig. 4b depicts the strain gauge arrangement. The axial force load, followed by the internal pressure load, was 0–147 s. Accordingly, during axial force loading, the pipeline primarily experienced axial tensile deformation, while the circumferential measuring points R1-R12 generated compressive strain via Poisson's action. After the axial force loading was completed, internal pressure was applied, and the circumferential strain increased significantly. Figure 3a shows that the initial stage of internal pressure loading (147–600 s) ranged from 0 MPa to 3.4 MPa. The strain growth rate in R1-R12 was roughly the same, and the pipeline expanded uniformly. During the later loading stage, as the internal pressure exceeded 3.4 MPa, the strain change rate at each measuring point began to differ due to concentrated deformation at the failure site. The measuring point R11, located at the burst position, had the most significant strain growth rate and value. As the pipeline was about to burst approached failure, the strain increased linearly and peaked. Because of the influence of expansion and compression, the strain growth rate was slowest, and the strain was smallest at measuring points R2 and R3 adjacent to the failure site.

Figure 3b indicates that the axial and circumferential strain changes were entirely different. The pipeline's axial strain increased linearly during the axial force loading stage. At this point, the intact pipeline was in the elastic axial tension stage, and the strain increase rate at each measuring point was identical. An internal pressure load was then applied, focusing on the deformation of the intact pipeline under internal pressure in the circumferential strain. The axial strain remained at a plateau stage, almost unchanged.

After 1300 s of loading and an internal pressure greater than 9.05 MPa, the intact pipeline's deformation concentrated at the burst location. The axial strains at measuring points R15 and R18, which were close to the blasting position, began to rise due to compression at the failure site and boundary constraints, with the strain being the maximum among all measuring points. The axial strain at measuring point R17 began to decrease due to the expansion of the failure site, the lowest value among all measuring points.

**Verification of the calculation equation**
Table 2 compares the error distributions of the 35 burst tests to demonstrate the superiority of the limit state equation in estimating failure pressure. The 35 experiments include one experiment

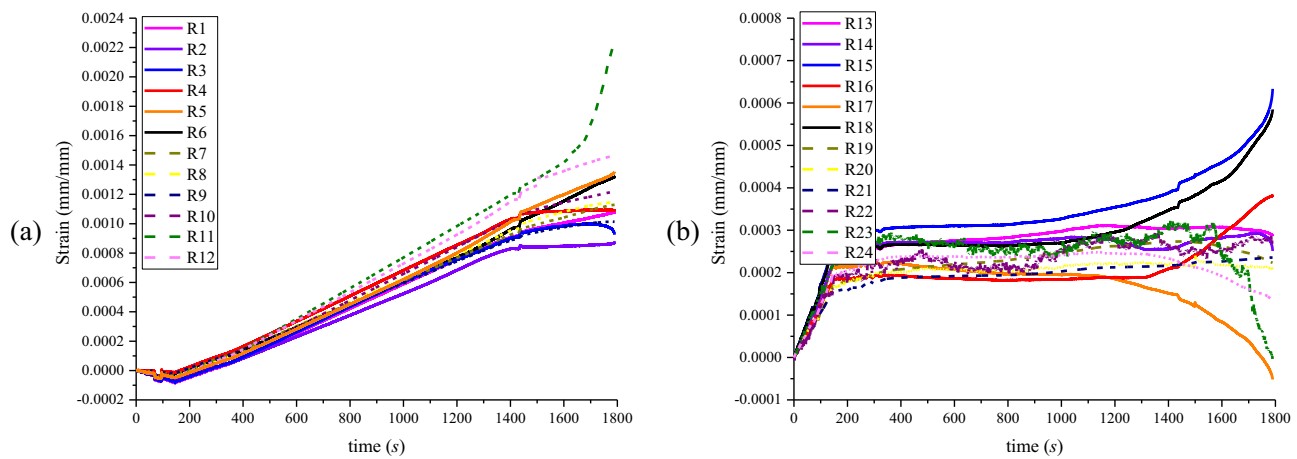

**Fig. 3 | Strain curve of an intact pipeline. a** Time series variation curves for circumferential strain at 12 measurement points (R1-R12). The axial force load was 0–147 s, and the pipeline was primarily subjected to axial tensile deformation. The circumferential strain R1-R12 generated compressive strain via Poisson's action. After the axial force loading was completed, internal pressure was applied, which significantly increased the circumferential strain. **b** Time series variation curves for axial strain at 12 measuring points (R13-R24). The pipeline's axial strain increased linearly during the axial force loading stage. The pipeline primarily deformed along its circumference during the internal pressure loading stage. The axial strain remained stable, with a small amplitude of variation. As the internal pressure exceeded 9.05 MPa, the pipeline's deformation concentrated at the burst position. With the expansion, the axial strain at measuring points R15 and R18 near the burst location increased. The measuring point R17 at the edge of the expansion zone significantly reduced axial strain with compression.

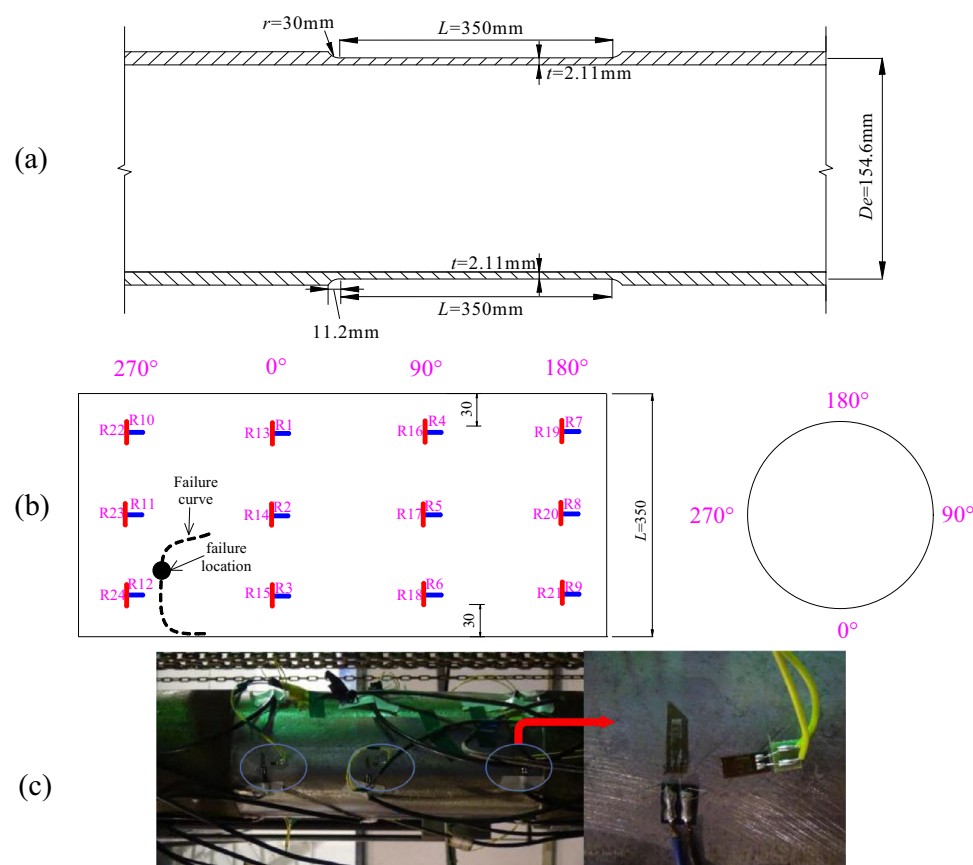

**Fig. 4 | The test pipeline and the strain gauge. a** A schematic diagram of the test pipeline (mm). The average diameter $D_e$ of the test pipeline was 154.6 mm, the average wall thickness $t$ was 2.11 mm, and the length $L$ was 350 mm. The experimental pipeline ended with a circular arc transition with a radius of 30 mm and a length of 11.2 mm to prevent stress concentration. **b** Strain gauge layout in an intact pipeline section (mm). The blue marks are strain gauges for circumferential strain, while the red marks are for axial strain. At the end and middle of the experimental pipeline, three layers of strain gauges are arranged in circumferential directions of 0°, 90°, 180°, and 270°, respectively. Therefore, strain gauges were placed in 12 different locations along the pipeline, with each location divided into axial and circumferential strain gauges. The failure occurred near the bottom of the experimental pipeline, between 270° and 0° (360°). **c** Physical diagram of the strain gauge.

illustrated in Figs. 4, 5 experiments from ref. 51, and 29 experiments from ref. 52. Specifically, it compares the error parameters of this study's proposed evaluation method with various yield criteria. It shows that the von Mises criterion was the most adaptable and stable of the four conventional yield criteria, with an average error of only 5.26%. Moreover, the standard deviation was only 0.0429, the lowest value among the four yield criteria. The Tresca yield criterion had the worst adaptability, with the highest average error and a wide range of prediction results. Accordingly, as analyzing burst failure of pipeline with complex loads, the second principal stress must be considered. The burst failure of pipelines carrying complex loads results from the first and second principal stresses acting simultaneously. In this sense, the yield criteria ASSY and TS had similar average errors and standard deviations, with ASSY (standard deviation = 0.0580) exhibiting a better applicability. The ASSY prediction results were conservative, whereas those of TS were dangerous. Between the ASSY and TS yield criteria, factor $b$ in the von Mises criterion determined the extent to which the second principal stress influenced the failure stress. The von Mises criterion was more closely related to the characteristics of pipeline blasting failure under complex loads, hence the prediction results were more accurate.

To compensate for the limitations of the experimental data, FE results were used to validate the developed calculation equation. Table 3 shows the errors of the FE models. As shown in the said table, the average error of the Tresca yield criteria was lower than in Table 2. This was due to a smaller axial force ($\frac{\sum(\sigma_L)}{\sigma_u} = 0.3$) and a lower impact

level of the second principal stress, which reduced the Tresca criterion's error. At the same time, the error in the TS criteria increased. The average error and standard deviation for ASSY were the smallest. Meanwhile, the Tresca and von Mises yield criteria differed only slightly from the ASSY criteria, and they all exhibited good adaptability.

According to the overall distribution of errors (Tables 2 and 3), the von Mises yield criterion presented the greatest adaptability, with the lowest error (7.90%) and standard deviation (0.0955) of any yield criterion. The ASSY yield criterion exhibited the next highest applicability, while there were significant errors in the Tresca and TS yield criteria (17–18%).

## Discussion

The UYC and the three-dimensional mechanical model proposed a model for assessing the burst pressure of pipelines under complex loads. The full-scale burst test clarified the pipeline's failure mode under complex loads and the applicability of various yield criteria in failure pressure assessment. The conclusions are as follows.

(1) Since the magnitudes of the three principal stresses with complex loads differed, pipeline limit state equations likewise differed.

(2) Circumferential stress remained a key indicator of internal pressure failure in pipelines carrying complex loads. The maximum circumferential failure strain was roughly 3.43 times the maximum axial failure strain.

(3) This study developed a relatively accurate method for determining failure pressure based on principal stress distribution under complex loads. The von Mises yield criterion, followed by the ASSY yield criterion, exhibited good applicability in a wide range of load combinations.

## Methods

This study discusses pipelines' bearing capacity under complex loads through theoretical derivation, experimentation, numerical simulation, and data analysis. First, it developed the pipeline's limit state equation using the three-dimensional stress distribution model. This equation represents a set of different yield criteria. The self-developed "Complex Load Testing Machine for Pipelines" was used to conduct failure experiments on pipelines with internal pressure and axial force. Then, this study determined the failure mode and the pipeline's internal pressure ultimate bearing capacity using the temporal variation characteristics of strain at various pipeline positions. Finally, a failure pressure database was created using the experiments described

**Table 2 | The errors of each proposed evaluation method (burst test)**

| Index | Error with different yield criterias | | | |
|---|---|---|---|---|
| | Tresca ($b=0$) | ASSY ($b=0.168$) | von Mises ($b=0.366$) | TS ($b=1$) |
| min | −0.11% | −0.11% | 0.16% | −0.27% |
| max | −59.85% | −23.90% | 16.93% | 35.60% |
| average | 12.60% | 7.83% | 5.26% | 7.80% |
| standard deviation | 0.1278 | 0.0580 | 0.0429 | 0.0841 |

error = $(p_f - p_T)/p_T \times 100\%$; $P_f$ is the predicted value; $P_T$ denotes the test's failure pressure; average = $\sum |\text{error}|/35$; $b$ is the yield criterion parameter.

**Table 3 | The errors of each proposed evaluation method (FE model)**

| steel grade | case | error with different yield criteria | | | |
|---|---|---|---|---|---|
| | | Tresca ($b=0$) | ASSY ($b=0.168$) | von Mises ($b=0.366$) | TS ($b=1$) |
| X52 | C-C0.3-B0.3 | −0.01% | 6.16% | 10.38% | 16.38% |
| | C-C0.3-B0.05 | 14.27% | 16.93% | 19.58% | 25.72% |
| | C-T0.3-B0.3 | −14.00% | −7.37% | −0.73% | 14.59% |
| | C-T0.3-B0.05 | 18.30% | 8.41% | 1.65% | −7.97% |
| | T-C0.3-B0.3 | −6.37% | 0.89% | 8.15% | 24.92% |
| | T-C0.3-B0.05 | 9.05% | 13.26% | 17.47% | 27.20% |
| | T-T0.3-B0.3 | −6.37% | 5.43% | −9.23% | −12.53% |
| | T-T0.3-B0.05 | 9.04% | 13.41% | 17.48% | 27.85% |
| min | | −14.00% | −7.37% | −9.23% | −12.53% |
| max | | 18.30% | 16.93% | 19.58% | 27.85% |
| average | | 9.68% | 8.98% | 10.58% | 19.65% |
| Standard deviation | | 0.1138 | 0.0781 | 0.1023 | 0.1609 |

Error = $(p_f - p_T)/p_T \times 100\%$; $P_f$ is the predicted value; $P_T$ denotes the test result; average = $\sum |\text{error}|/8$; $b$ is the yield criterion parameter.

in this study and other experimental and numerical model results, and the applicability of various yield criteria were compared and analyzed.

## Theory of the limit state equation

The UYC[53–55] considers the contribution of the second principal stress $\sigma_2$ to structural failure. Steel pipelines can be considered a tension-compression isotropic material. The UYC criteria may be expressed as follows:

$$C = \begin{cases} \tau_{13} + b\tau_{12}, & \tau_{12} \geq \tau_{23} \\ \tau_{13} + b\tau_{23}, & \tau_{12} < \tau_{23} \end{cases} \tag{3}$$

where $\tau_{13}$, $\tau_{12}$, and $\tau_{23}$ are the principal shear stresses, and $C$ is a material parameter.

$$C = \frac{1+b}{2}\sigma_{UE} \tag{4}$$

where $\sigma_{UE}$ is the UYC equivalent stress.

From Eqs. (3) and (4):

$$\sigma_{UE} = \begin{cases} \sigma_1 - \frac{1}{1+b}(b\sigma_2 + \sigma_3), & \sigma_2 \leq \frac{\sigma_1+\sigma_3}{2} \\ \frac{1}{1+b}(\sigma_1 + b\sigma_2) - \sigma_3, & \sigma_2 > \frac{\sigma_1+\sigma_3}{2} \end{cases} \tag{5}$$

where $\sigma_1$, $\sigma_2$, and $\sigma_3$ are the principal stresses, and $\sigma_1 \geq \sigma_2 \geq \sigma_3$.

The UYC is a set of various criteria, as illustrated in Supplementary Fig. 1[56]. The value range of $b$ is $0 \leq b \leq 1$.

(1) As $b = 0$, the effect of the second principal shear stress on the equivalent stress of plastic failure of the pipeline is completely ignored, and the criterion is analogous to the Tresca yield criterion.

(2) As $b = 1$, the equivalent stress of plastic failure in a pipeline is influenced by the first and second principal shear stresses, and the weights of the two stresses are equal. The yield criterion is based on twin shear stress (TS).

(3) As $0 < b < 1$, the second principal shear stress affects the equivalent stress of plastic failure in a pipeline, but its weight is lower than the first principal shear stress. For example, as $b = 1/(1+\sqrt{3}) \approx 0.366$, the criterion approximates to the von Mises criterion; as $b = (8\sqrt{3} - 10)/23 \approx 0.168$, the criterion is the average shear stress yield criterion (ASSY).

For thin-walled pipelines ($(D_e/t) \geq 20$), Eqs. (6) and (7) can be used to calculate circumferential stress $\sigma_h$ and axial stress $(\sigma_L)_p$ by pressure. Radial stress $\sigma_r$ can be ignored relative to the other stresses.

$$\sigma_h = p\frac{D_e}{2t} \tag{6}$$

$$(\sigma_L)_p = \frac{\sigma_h}{2} \tag{7}$$

where $D_e$ is the outer diameter.

Equations (8) and (9) can be used to calculate the axial stress $(\sigma_L)_{\Delta T}$ or $(\sigma_L)_{Nc}$ generated by the temperature difference $\Delta T$ or the longitudinal force $N_c$.

$$(\sigma_L)\Delta T = -E\lambda\Delta T \tag{8}$$

$$(\sigma_L)_{Nc} = \frac{N_C}{A} \tag{9}$$

where $E$ is the elastic modulus (MPa), $\lambda$ denotes the temperature expansion coefficient (/°C), $\Delta T$ represents the temperature difference (°C), $A$ is the circumferential area of the pipeline (mm²) in which $A = \frac{\pi}{4}(D_e^2 - D_i^2)$, and $D_i$ denotes the inner diameter.

The axial stress $(\sigma_L)_{Mb}$ generated by the bending moment $M_b$ is given by Eq. (10).

$$(\sigma_L)_{Mb} = \pm\frac{M_b}{I}\frac{D_e}{2} \tag{10}$$

where $I$ is cross sectional moment of inertia in which $I = \frac{\pi}{64}(D_e^4 - D_i^4)$.

The total axial stress $\sigma_L$ is the sum of $(\sigma_L)_p$ and the stress $\Sigma(\sigma_L)$ generated by bending moment, axial force, and others.

$$\sigma L = (\sigma L)p \pm \sum(\sigma L) \tag{11}$$

For the burst failure of a pipeline, $\sigma_{UE}$ can be taken as $\sigma_u$[57]. The properties of axial stress allow the solution of the limit state equation in the following situations:

1. As the additional loadings are positive (tensile), and $\sigma_1 = \sigma_L$, $\sigma_2 = \sigma_h$, $\sigma_3 = 0$.

(1) If $\sigma_2 \leq \frac{\sigma_1+\sigma_3}{2}$, according to Eq. (5):

$$\sigma_{UE} = \sigma_1 - \frac{1}{1+b}(b\sigma_2 + \sigma_3) = \sigma_u \tag{12}$$

$$\sigma 1 = (\sigma L)p + \Sigma(\sigma L) \tag{13}$$

$$\sigma_u = p_0\frac{D_e}{2t} \tag{14}$$

where $p_0$ is the intact pipeline's burst pressure.

Substituting Eqs. (13) and (6) into Eq. (12):

$$(\sigma_L)_p + \Sigma(\sigma_L) - \frac{b}{1+b}*p\frac{D_e}{2t} = \sigma_u \tag{15}$$

$$\frac{(\sigma_L)_p}{\sigma_u} + \frac{\Sigma(\sigma_L)}{\sigma_u} - \frac{b}{1+b}\frac{p\frac{D_e}{2t}}{\sigma_u} = 1 \tag{16}$$

Substituting Eqs. (6), (7), and (14) into Eq. (16):

$$\frac{\frac{1}{2}*p\frac{D_e}{2t}}{p_0\frac{D_e}{2t}} + \frac{\sum(\sigma_L)}{\sigma_u} - \frac{b}{1+b}\frac{p\frac{D_e}{2t}}{p_0\frac{D_e}{2t}} = 1 \tag{17}$$

In this case, the limit state equation is defined as:

$$\left(\frac{1}{2} - \frac{b}{1+b}\right)\frac{p}{p_0} + \frac{\sum(\sigma_L)}{\sigma_u} = 1 \tag{18}$$

(2) If $\sigma_2 > \frac{\sigma_1+\sigma_3}{2}$, according to Eq. (5):

$$\sigma_{UE} = \frac{1}{1+b}(\sigma_1 + b\sigma_2) - \sigma_3 = \sigma_u \tag{19}$$

Substituting Eqs. (13) and (6) into Eq. (19):

$$\frac{1}{1+b}\left((\sigma_L)_p + \Sigma(\sigma_L) + b*p\frac{D_e}{2t}\right) = \sigma_u \tag{20}$$

$$\frac{1}{1+b}\left(\frac{(\sigma_L)_p}{\sigma_u} + \frac{\sum(\sigma_L)}{\sigma_u} + b*\frac{p\frac{D_e}{2t}}{\sigma_u}\right) = 1 \tag{21}$$

Substituting Eqs. (6), (7), and (14) into Eq. (21):

$$\frac{1}{1+b}\left(\frac{\frac{1}{2}*p\frac{D_e}{2t}}{p_0\frac{D_e}{2t}} + \frac{\sum(\sigma_L)}{\sigma_u} + b*\frac{p\frac{D_e}{2t}}{p_0\frac{D_e}{2t}}\right) = 1 \tag{22}$$

**Table 4 | Data of burst tests**

| literature Sources | test No. | diameter (mm) | grade | D/t | axial stress (MPa) | yield strength (MPa) | tensile Strength (MPa) |
|---|---|---|---|---|---|---|---|
| Paslay et al.[53] | 13 | 88.9 | L80 | 14.5 | 348.79247 | 695.201 | 753.766 |
| | 16 | 88.9 | L80 | 18.0 | 586.66283 | 695.201 | 753.766 |
| | 17 | 88.9 | L80 | 18.6 | 638.14491 | 695.201 | 753.766 |
| | 18 | 177.8 | K55 | 26.7 | 454.31971 | 465.764 | 737.23 |
| | 19 | 196.85 | Q125 | 18.1 | 884.73801 | 905.346 | 993.538 |
| Lasebikan et al.[54a] | 22 °C–1 | 8 | 125 | | 450 | 969 | 1063 |
| | 22 °C–2 | | | | 450 | | |
| | 22 °C–3 | | | | 460 | | |
| | 22 °C–4 | | | | 550 | | |
| | 22 °C–5 | | | | 540 | | |
| | 22 °C–6 | | | | 650 | | |
| | 22 °C–7 | | | | 655 | | |
| | 22 °C–8 | | | | 665 | | |
| | 22 °C–9 | | | | 750 | | |
| | 22 °C–10 | | | | 751 | | |
| | 22 °C–11 | | | | 740 | | |
| | 22 °C–12 | | | | 750 | | |
| | 90 °C–1 | | | | 400 | 881 | 948 |
| | 90 °C–2 | | | | 410 | | |
| | 90 °C–3 | | | | 410 | | |
| | 90 °C–4 | | | | 550 | | |
| | 90 °C–5 | | | | 551 | | |
| | 90 °C–6 | | | | 650 | | |
| | 110 °C–1 | | | | 400 | 851 | 935 |
| | 110 °C–2 | | | | 400 | | |
| | 110 °C–3 | | | | 400 | | |
| | 110 °C–4 | | | | 550 | | |
| | 110 °C–5 | | | | 545 | | |
| | 110 °C–6 | | | | 550 | | |
| | 160 °C–1 | | | | 400 | 821 | 888 |
| | 160 °C–2 | | | | 400 | | |
| | 160 °C–3 | | | | 550 | | |
| | 160 °C–4 | | | | 550 | | |
| | 160 °C–5 | | | | 650 | | |

[a]Note: Data were obtained from the reference literature's charts; $D$ and $t$ represent the diameter and the wall thickness, respectively.

In this case, the limit state equation is defined as:

$$\left(\frac{1}{2(1+b)} + \frac{b}{b+1}\right)\frac{p}{p_0} + \frac{1}{1+b}\frac{\sum(\sigma_L)}{\sigma_u} = 1 \tag{23}$$

2.  As the additional loadings are positive (tensile), and $\sigma_1 = \sigma_h$, $\sigma_2 = \sigma_L$, $\sigma_3 = 0$, i.e., $\sigma_2 > \frac{\sigma_1+\sigma_3}{2}$, the limit state equation is defined as:

$$\frac{1}{1+b}\left(1+\frac{b}{2}\right)\frac{p}{p_0} + \frac{b}{1+b}\frac{\sum(\sigma_L)}{\sigma_u} = 1 \tag{24}$$

3.  As the additional loadings are negative (compressive), and $\sigma_1 = \sigma_h$, $\sigma_2 = 0$, $\sigma_3 = \sigma_L$.
(1)  If $\sigma_2 \leq \frac{\sigma_1+\sigma_3}{2}$

$$\sigma L = (\sigma L)p - \sum(\sigma L) \tag{25}$$

In this case, the limit state equation is defined as:

$$\left(1 - \frac{1}{2(1+b)}\right)\frac{p}{p_0} - \frac{1}{1+b}\frac{\sum(\sigma_L)}{\sigma_u} = 1 \tag{26}$$

(2)  If $\sigma_2 > \frac{\sigma_1+\sigma_3}{2}$

The limit state equation is defined as:

$$\left(\frac{1}{b+1} - \frac{1}{2}\right)\frac{p}{p_0} + \frac{\sum(\sigma_L)}{\sigma_u} = 1 \tag{27}$$

4.  As the additional loadings are negative (compressive), and $\sigma_1 = \sigma_h$, $\sigma_2 = \sigma_L$, $\sigma_3 = 0$, i.e., $\sigma_2 \leq \frac{\sigma_1+\sigma_3}{2}$, the limit state equation is defined as:

$$\left(1 - \frac{b}{2(1+b)}\right)\frac{p}{p_0} + \frac{b}{1+b}\frac{\sum(\sigma_L)}{\sigma_u} = 1 \tag{28}$$

## Full-scale burst test

A full-scale test with axial force and internal pressure was carried out to examine the burst model of intact pipelines under various loads. Supplementary Figs. 2 and 3 show the processing of experimental pipelines. Supplementary Fig. 4 and Supplementary Note 1 describe the experimental setup and procedure. The overall wall thickness reduction treatment was performed in the middle of the pipeline to produce an intact, defect-free section. Circular arcs cross both processed and unprocessed parts to prevent stress concentration. Figure 4a depicts the axial interface of the processed pipeline.

The transition arc had a radius of 30 mm and a length of 11.2 mm. The pipeline's strength grade was Q235, a Chinese standard steel grade. It is a carbon structural steel with a yield strength standard of 235 MPa. Q235 steel is widely used in construction, bridges, pipelines, and other structural applications because of its good weldability, machinability, and strength. The experimental pipeline's measured yield and tensile strength were 280 and 415.5 MPa, respectively.

The primary cause of axial force during the operation of buried pipelines, among other factors, is temperature differential. In this study, the temperature difference was set to 27 °C. Because temperature-induced axial force is challenging to release via pipeline axial elongation, the temperature difference causes the pipeline to generate axial force, which can be estimated using the following equation:

$$N_c = -AE\lambda\Delta T \tag{29}$$

where $N_c$ is a negative value and represents the axial compression force (N). The calculated value of $N_c$ was 68 kN, which was used in the experiment.

Fig. 4b, c present the arrangement mode of the strain gauge for the burst test of intact pipelines. The strain gauges were arranged in four directions along the pipeline's axial circular section, namely 0°, 90°, 180°, and 270°. Each direction had circumferential and axial strains at both ends and the middle to monitor the strain in the intact pipeline.

## Data analysis

Table 4 displays the full-scale burst test data from refs. 51,52, which included axial force and internal pressure. Supplementary Table 1 shows detailed predicted values and error data.

## Numerical simulation

The FE model was chosen for validation under the following criteria:

(1) The burst test lacked data at $0.2 < \frac{\sum(\sigma_L)}{\sigma_u} < 0.4$; thus, the value of $\frac{\sum(\sigma_L)}{\sigma_u}$ was taken as 0.3.

(2) The burst test lacked axial compression data; thus, axial force data was supplemented with compressive stress.

(3) The experiment lacked bending moment load; thus, pipeline burst data with "bending moment-axial force-internal pressure" was added.

In this case, ABAQUS created a geometric model, and the corresponding finite element model was built with the three-dimensional solid unit C3D8R. The pipeline was divided into four layers of units in the thickness direction, 48 units in the circumferential direction, and 88 units in the axial direction. Reference points were established at both pipeline ends, and rigid beam constraints were utilized to connect the reference points to the pipeline end nodes. The analyzed model is depicted in Supplementary Fig. 5.

The bending moment ($M_b$) and the axial force ($N_c$) were applied to the designated reference node while gradually increasing the internal pressure on the inner surface nodes until the pipe burst.

The nonlinear arc-length method algorithm was utilized to solve the finite element model. The simulation applied a pipeline material with isotropic hardening plasticity. This study used the well-known Ramberg-Osgood model to represent the stress-strain relationship accurately.

The burst data in Table 4 was used to validate the model, and the results are shown in Table 5. The average error was 3.11%, which falls within the acceptable range. Thus, the model's accuracy was verified.

**Table 5 | The errors of the FE model**

| steel grade | case | failure pressure in the test (MPa) | failure pressure with FE model (MPa) | error |
|---|---|---|---|---|
| Q235 | test in the paper | 11.69 | 11.70 | 0.09% |
| L80 | 13 | 115.48 | 111.67 | −3.30% |
| L80 | 16 | 64.08 | 68.68 | 7.18% |
| L80 | 17 | 50.64 | 50.66 | 0.03% |
| K55 | 18 | 54.84 | 57.37 | 4.62% |
| Q125 | 19 | 63.39 | 61.20 | −3.45% |
| min | | −3.45% | | |
| max | | 7.18% | | |
| average | | 3.11% | | |
| Standard deviation | | 0.0427 | | |

Error = $(p_{FE} − p_T)/p_T \times 100\%$; $P_{FE}$ is the predicted value; $P_T$ denotes the failure pressure of the test; average = $\sum |error|/6$. The "residual wall thickness stress criterion" determined the failure pressure of the pipelines[6,59].

**Table 6 | Results of finite element model**

| test No. | diameter (mm) | grade | yield strength (MPa) | tensile Strength (MPa) | t | axial Stress (MPa) | bend moment (N·mm) | failure pressure (MPa) |
|---|---|---|---|---|---|---|---|---|
| C-C0.3-B0.3 | 914.4 | X52 | 413 | 545 | 9.525 | −0.3σ_u | 0.3M_O | 9.11 |
| C-C0.3-B0.05 | | | | | | −0.3σ_u | 0.05M_O | 9.94 |
| C-T0.3-B0.3 | | | | | | 0.3σ_u | 0.3M_O | 13.20 |
| C-T0.3-B0.05 | | | | | | 0.3σ_u | 0.05M_O | 14.39 |
| T-C0.3-B0.3 | | | | | | −0.3σ_u | 0.3M_O | 12.13 |
| T-C0.3-B0.05 | | | | | | −0.3σ_u | 0.05M_O | 10.41 |
| T-T0.3-B0.3 | | | | | | 0.3σ_u | 0.3M_O | 12.13 |
| T-T0.3-B0.05 | | | | | | 0.3σ_u | 0.05M_O | 10.41 |

"-" represents the compressive stress; $M_O$ is the ultimate elastic buckling moment, in which $M_O = D^2 t\sigma_y$[60,61]; "C/T-C0.3/T0.3-B0.3/B0.05" means "failure pressure on the **C**ompression side or **T**ensile side caused by bending moment," "**C**ompression or **T**ensile stress is **0.3**$\sigma_u$" and "**B**ending moment is **0.3**$M_O$ or **0.05**$M_O$."

The size and strength data of X52 from the Yi Shuai and Xiao Zhang models were used for analysis[58]. The reasons for utilizing the said data are as follows:

(1) In the absence of medium-strength pipeline data in the burst test, X52 steel was chosen as a representative sample for analysis.

(2) Due to a lack of data on the large diameter-to-thickness ratio in the test (the burst test in Table 3 has a ratio of 14.5–73.0), $D/t = 96$ was chosen as a representative value. Table 6 shows the specific FE model.

## Data availability

Source data is available as Source Data file. It also has been deposited in the Zenodo database at https://doi.org/10.5281/zenodo.11118137. Source data are provided with this paper.

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

## Acknowledgements

This project is supported by Research Projects in Henan Province (23A560013), the National Key R&D Program of the "14th Five-Year Plan" (2022YFC3801000), the Henan Provincial Youth Science Foundation (232300421328), the Excellent Youth Innovation Research Group Project (242300421001), and the Henan Province University Science and Technology Innovation Team (23IRTSTHN004).

## Author contributions

S.M. performed data analysis, developed finite element models, solved theoretical solutions to limit state equations, and wrote the manuscript. F.H., W.N., and D.X. conducted experimental operations and data collection. Z.H. arranged the manuscript's language. Z.K. carried out drawing-related tasks.

## Competing interests

The authors declare no competing interests.
