## [Peer Review File · Nature Communications]

Limit state equation and failure pressure prediction model of pipeline with complex loadingEditorial Note: Parts of this Peer Review File have been redacted as indicated to remove third-party material where no permission to publish could be obtained.

REVIEWER COMMENTS

Reviewer #1 (Remarks to the Author):

The authors intended to establish the model to evaluate the limit state of pipes under complex loading (internal pressure, axial force and bending moment). The authors provided the limit state equations based on the several yield criteria and conducted the burst experiment as well to examine the predicted failure pressure by proposed equation. I appreciate the authors contribution to this field.

I think the purpose of the research is good; however, the manuscript includes some unclear or conflicted descriptions about the authors' work. In addition to it, the detail of the FE model is not provided in Sec.5.2 (see the comment item (14) in the attached file regarding this point). Considering these points, I judge the manuscript should be major revised.

For the detailed comments, please check the attached file.

Reviewer #2 (Remarks to the Author):

1.What are the main innovations of this manuscript? Many failure pressure evaluation models are proposed every year. What is special about the evaluation model in this manuscript?

2.In abstract, "The current evaluation method for failure pressure of steel pipelines does not consider the impact of other loads besides internal pressure." As far as I know, many scholars have studied the failure pressure of pipelines under complex loads. Such as Shuai Y, Zhang X, Feng C, et al. A novel model for prediction of burst capacity of corroded pipelines subjected to combined loads of bending moment and axial compression[J]. International Journal of Pressure Vessels and Piping, 2022, 196: 104621. Chen Z F, Wang W, Yang H, et al. On the effect of long corrosion defect and axial tension on the burst pressure of subsea pipelines[J]. Applied Ocean Research, 2021, 111: 102637. Personally, I don't think that's accurate enough.

3.There is considerable debate as to which is the more accurate criterion for the strength of metallic materials under complex loads. Which strength criterion is accurate may require extensive experimentation to give a suitable conclusion. There is less experimental data in this paper and it is recommended to find some experimental data from the literature to fully validate the conclusion.

4.How are Equation 15 and Equation 17 obtained from Equation 6? Please explain.

5.In section 4, what effect does the loading sequence have on the experimental results?

6.In section 5.2, it is best to introduce the finite element model in detail, such as modeling, mesh division, boundary conditions, etc.

7.In conclusions, "(2) Circumferential stress remains a key indicator of internal pressure failure in pipelines with complex loads. Tensile stress will increase the internal pressure bearing capacity of the pipeline, while compressive stress has the opposite effect" Conclusion 2 conflicts with the experimental data in the literature (Lasebikan B A, Akisanya A R. Burst pressure of super duplex stainless steel pipes subject to combined axial tension, internal pressure and elevated temperature[J]. International Journal of Pressure Vessels and Piping, 2014, 119: 62-68.) According to Lasebikan's experiments, axial tension, regardless of the value, will reduce the failure pressure of the pipe. Please check.

Paper ID: NCOMMS-23-43457

Paper Title: Research on the Limit State Equation and Failure Pressure Prediction Model of Pipeline with Complex Loading

[Comments]

The following are the specific comments for the paper. Revise the manuscript as necessary.

- (1) The authors often use the expression “perfect pipeline”. The meaning of this term is not clear. If the authors intend to refer to the pipe without defect such as wall thinning, “intact pipe” or “pipe without defects” would be more accurate and easier to understand.
- (2) In “Introduction” section, some references are not correctly cited. For example, at the eighth line in page 4, the reference is cited as “Vivianne and Nadege et al. (Bruere et al., 2019)” at). The authors’ names do not match the following reference. Check the whole manuscript to cite the references correctly.
- (3) Some abbreviations are not explained; for example, “JIP” at the fourth line in page 3.
- (4) Some parameters used in the equations are not explained (for example, D_e , D_i , α in Eqs. (6) ~ (9)). Most of them are understandable; however, they should be determined clearly in the manuscript. In addition, in Eq. (23), it seems that the authors use λ instead of α . Use the same parameter through the manuscript.
- (5) Page 10, Fig.2 and Table 2; 1) Dimensions in Fig.2 and those listed in Table 2 are different. 2) In Fig.2, the characters to describe the transition length is too small. 3) In Fig.2, add the unit of length. 4) In Table 2, are “yield stress” and “tensile stress” the value prescribed in the standard or actual data from the inspection certificate report of actual pipe used in the experiment?
- (6) Page 10, line 7; it is better to add a short explanation what the standard “Q235” is based on.
- (7) Page 12, 4.3; 1) Section title; “The process of load loading” -> “The process of loading”. 2) It is helpful for the readers to provide some schematic illustrations to explain the experimental setup. 3) In the procedure (3), the authors explain that they “inject water”; I thought that the water injection finished in the procedure (1). Didn’t it? Please clarify the procedure. 4) I cannot understand well the procedure (4). The authors describe that the force-controlled loading is used with a loading rate of 0.25 MPa/min. Does “0.25 MPa” mean the internal pressure? After the yield stage, the authors explain that they conduct displacement-controlled test by a loading rate of 1.5 mm/min. What displacement is used for this control?
- (8) Page 13, Fig.5; 1) Add the explanation that the green marks are the strain gage for hoop strain, and the red marks are those for axial strain. 2) Prepend “R” to numbers those represent the strain measurement points. (so as to correspond with the description in the text).
- (9) Page 14, the fourth line in the section 4.4.3, “Considering the vertical placement and the difference of wall thickness”; 1) I cannot understand well the meaning of “vertical placement”. 2) Did the authors measure the actual wall thickness distribution? If the authors have information about the wall thickness, it is better to discuss the influence of wall thickness on the rupture failure with such evidence.

- (10) Page 14, Fig.6; please clarify the difference between those four photos.
- (11) Page 15; to understand the discussion in this section, it is helpful to show where the failure location is in the strain measurement map (Fig.5(a)).
- (12) Page 17, Table 3; add units for yield strength, tensile strength, and failure pressure.
- (13) Page 18, Table 4; please provide the predicted pressure values by each prediction method. To notify again the actual failure pressure values in the tests in Table 4 is helpful.
- (14) From Page 18, 5.2 The FE model; 1) provide the detail of FE analysis model. There are no description what kind of the FE code is used, what the FE model configuration is, what kind of elements are used, how the model is meshed, how the material properties are modeled, and

so on. 2) the authors explained as $\frac{\sum(\sigma_L)}{\sigma_u}$ is taken as 0.25; however, in Table 5 and the related

explanations, the authors describe that the value is set as 0.3. Which of the condition is correct?

3) in the note of Table 5, the authors describe “Failure pressure on the compression side or tensile side caused by bending moment”. I cannot understand well the meaning of this sentence.

4) Why the dimensions and the steel grade of pipes in the FE analysis are different from the experiments the authors conducted? Did the authors compare the experimental results with the FE analytical results? If not, how did the authors evaluate the reliability of the FE analysis model?

Date: Dec. 25, 2023

To: Nature Communications

From: SUN Ming-ming, FANG Hong-yuan*, WANG Nian-nian, DU Xue-ming, Zhai Ke-Jie

Re: Manuscript No.: NCOMMS-23-43457

Title: Research on the Limit State Equation and Failure Pressure Prediction Model of Pipeline with Complex Loading

Thank you for taking the time to review our submission. Your Comments are invaluable and significant to improve our paper. We have carried out all the corrections as suggested. The Comments for this paper and the response to these comments are as following. The revised content in the manuscript has been highlighted in yellow.

Reviewer #1

1. Comment: (1) The authors often use the expression “perfect pipeline”. The meaning of this term is not clear. If the authors intend to refer to the pipe without defect such as wall thinning, “intact pipe” or “pipe without defects” would be more accurate and easier to understand.

Author's Response:

Through analysis of relevant literature, the expression of pipelines without defects is "intact pipeline", "pipe without defects" or "defect-free pipeline". Therefore, the "perfect pipeline" in the manuscript is replaced with "intact pipeline". The modified content is as follows:

“Based on the three-dimensional stress model of pipelines with complex loads, a failure pressure assessment model for intact pipelines has been proposed in the paper.”

“ Mechanical model analysis and bearing capacity evaluation of intact pipelines are the foundation of pipeline system integrity evaluation(Sun et al., 2022).”

“This equation can evaluate the failure pressure of intact pipelines with multiple or individual external loads (bending moment, axial compressive force, axial tensile force, and internal pressure).”

“Where p_0 is the internal pressure bearing capacity of the intact pipeline.”

“To analyzed the burst model of intact pipelines with different loads, full-scale test was conducted with axial force and internal pressure. The characteristics of intact pipeline burst test are as follows: (1) Unable to predict the failure location and provide effective damage protection; (2) The constraints at both ends of the pipeline are prone to stress concentration and damage, which cannot accurately reflect the failure mode of the intact pipeline.”

“Therefore, in this test, the overall wall thickness reduction treatment is carried out in the middle of the pipeline to produce a section of intact pipeline without defects.”

“Fig. 5 shows the arrangement mode of strain gauge for the burst test of intact pipelines. The strain gauges are respectively arranged in four directions of 0° , 90° , 180° and 270° of the axial circular section of the pipeline. Each direction is arranged with circumferential and axial strains at both ends and middle position to monitor the strain in the intact pipeline.”

“**Fig. 5** Layout of strain gauge in intact pipeline section”

“Fig. 6 shows photos of the explosion failure of intact pipelines.”

“**Fig. 6** Failure modes of intact pipelines”

“Fig. 7 shows the loading curve of a intact pipeline”

“**Fig. 7** The loading curve of intact pipeline”

“Afterwards, the internal pressure load was applied, and the deformation of the intact pipeline with internal pressure was mainly concentrated in the circumferential strain.”

“**Fig. 8** Strain curve of intact pipeline”

“After loading for 1300s and with an internal pressure greater than 9.05MPa, the deformation of the intact pipeline begins to concentrate at the burst location.”

2. Comment: (2) In “Introduction” section, some references are not correctly cited. For example, at the eighth line in page 4, the reference is cited as “Vivianne and Nadege et al.

(Bruere et al., 2019)” at). The authors’ names do not match the following reference. Check the whole manuscript to cite the references correctly.

Author’s Response:

I deeply apologize for the inconvenience caused by the citation error and have made revisions based on the reviewer's comments. Simultaneously review the entire manuscript to avoid similar errors.

After verification, the citation error in the manuscript is due to an incorrect sorting of the author's name. The revised content is as follows:

“Bruère and Nadège et al. (Bruère et al., 2019) and this method can evaluate the impact of axial compressive force.

The FE method was adopted by Arumugam and Muhammad et al. (Arumugam et al., 2020) to analyzed the bearing capacity of colony corrosion defects with compressive load and internal pressure.

Plastic collapse assessment procedure was proposed by Shinji Konosu and Mukaimachi (Konosu and Mukaimachi, 2008) for pressurized vessel with local defect with bending moment. Mohd MH, Lee BJ et al. (Mohd et al., 2015) also investigate the same case of corroded pipeline.

Mondal Bipul and Dhar Ashutosh (Mondal and Dhar, 2019), Yong Bai and Soren Hauch (Bai and Hauch, 2001), Xiao Tian and Hong Zhang et al. (Tian et al., 2019) have also conducted similar studies.

Smith and Grigory (Smith and Grigory, 1996)proposed the global buckling failure envelopes of corroded pipeline accommodating combined stresses.”

3. Comment: (3) Some abbreviations are not explained; for example, “JIP” at the fourth line in page 3.

Author’s Response:

All the abbreviations have been explained and the revised content is as follows:

“Other publications of the published document (Bjornoy et al., 1997; BjØrnoy et al., 2001; Bjørnøy et al., 1999; Bjørnøy et al., 2000; BjØrnØy and Marley, 2001; Sigurdsson et al., 1999) refer to the "Reliability of corroded pipes" in the Joint Industry Projects (JIP) project developed by DET NORSKE VERITAS (DNV)”

“The criterion is twin shear stress yield criterion (TS).”

“The criterion is the average shear stress yield criterion (ASSY).”

4. Comment: (4) Some parameters used in the equations are not explained (for example, D_e , D_i , α in Eqs. (6) ~ (9)). Most of them are understandable; however, they should be determined clearly in the manuscript. In addition, in Eq. (23), it seems that the authors use λ instead of α . Use the same parameter through the manuscript..

Author’s Response:

I am very sorry for the inconvenience caused by writing mistake, the revisions have been made according to the reviewer's comment. Simultaneously review the entire manuscript to avoid similar errors.

The revised content is as follows:

“where D_e is the outer diameter of the pipeline.

$$(\sigma_L)_{\Delta T} = -E\lambda\Delta T \quad (8)$$

where E is the elastic modulus (MPa); λ is the temperature expansion coefficient ($^{\circ}\text{C}$); ΔT is the temperature difference ($^{\circ}\text{C}$), A is the circumferential cross-sectional area of the pipeline (mm^2) and $A = \frac{\pi}{2}(D_e^2 - D_i^2)$, D_i is the inner diameter.

where, I is cross sectional moment of inertia and $I = \frac{\pi}{64}(D_e^4 - D_i^4)$.

$$N_c = -AE\lambda\Delta T \quad (23)$$

where N_c is a negative value and represents the axial compression force (N).”

5. Comment: Page 10, Fig.2 and Table 2; 1) Dimensions in Fig.2 and those listed in Table 2 are different. 2) In Fig.2, the characters to describe the transition length is too small. 3) In

Fig.2, add the unit of length. 4) In Table 2, are “yield stress” and “tensile stress” the value prescribed in the standard or actual data from the inspection certificate report of actual pipe used in the experiment?

Author’s Response:

1) As Fig. 2 and Table 2 respectively represent the factory design values and measured average values of the pipeline, there is a difference in the data. The Fig.2 is unified as measured value.

2) The characters to describe the transition length is revised.

3) The unit of length has been added in Fig.2.

4) In order to accurately evaluate the bearing capacity of pipelines, the failure pressure assessment and table data in the manuscript are both actual data. The table 1 lists the test data.

Table 1 Tensile Test Results of Q235 Sample

Test case	σ_y (MPa)	σ_u (MPa)	σ_y/σ_u
Q235#1	284.2	412.0	0.690
Q235#2	275.7	419.1	0.658
Q235#3	280.0	415.5	0.674
min	275.7	419.1	0.658
max	284.2	419.1	0.690
average	280.0	415.5	0.674

6. Comment: (6) Page 10, line 7; it is better to add a short explanation what the standard “Q235” is based on.

Author’s Response:

A short explanation what the standard “Q235” is based on was added. The added content is as follows:

“Q235 is a Chinese standard steel grade, which is commonly used for structural purposes. It is

a carbon structural steel with a yield strength standard value of 235 MPa. Q235 steel is widely used in construction, bridges, pipelines, and other structural applications due to its good weldability, machinability, and strength properties.”

7. Comment: (7) Page 12, 4.3; 1) Section title; “The process of load loading” -> “The process of loading”. 2) It is helpful for the readers to provide some schematic illustrations to explain the experimental setup. 3) In the procedure (3), the authors explain that they “inject water”; I thought that the water injection finished in the procedure (1). Didn’t it? Please clarify the procedure. 4) I cannot understand well the procedure (4). The authors describe that the force-controlled loading is used with a loading rate of 0.25 MPa/min. Does “0.25 MPa” mean the internal pressure? After the yield stage, the authors explain that they conduct displacement-controlled test by a loading rate of 1.5 mm/min. What displacement is used for this control?.

Author’s Response:

- 1) The title has been revised in the manuscript.
- 2) Text and diagrams have been added to help readers understand the experimental setup.

The revised content is as follows:

“The pipeline burst test with internal pressure and axial force was conducted by applying a load through oil pressure, and the specific experimental setup is shown in Fig. 5. The 6000 kN hydraulics cylinder is installed inside the pipeline base and can apply an axial force of -6000kN-2000kN.

The internal pressure loading device includes water injection, drainage, and pressurization devices. The pressurization device is a set of stainless steel boosting cylinders driven by servo oil cylinders, with a stroke of 600 mm and the ability to apply a maximum pressure of 50MPa. The liquid in the water cylinder is squeezed through a pressurization device and enters the full-scale experimental pipeline through a water delivery pipe, and internal pressure is applied using the pressurization device. The control platform can control the pressure application method through sensors, including displacement control and load

(pressure) control.

(b) axial force loading system

(a) control platform

(c) internal pressure loading system

Fig. 5 Loading device and control platform for pipeline tests

”

3) The water injection of procedure (1) mainly fills the internal space of the pipeline through water injection and discharges air. The water injection of procedure (3) mainly involves injecting more water than the internal space of the pipeline into the pipeline to apply internal pressure. The main differences in water injection between procedure (1) and procedure (2) are reflected in the following three aspects:

- The water injection of procedure (1) is injected through tap water pressure, at which point the pipeline is not under internal pressure, while the water injection of procedure (3) is injected through a pressurization device, at which point the pipeline is under internal pressure;
- The purpose of water injection in procedure (1) is to exhaust the air inside the pipeline and fill the entire pipeline with water, while in procedure (3), water injection is to apply internal pressure to the test pipeline;

□ During the injection process of procedure (1), the drainage pipe is opened to exhaust the air inside the pipeline, while during the injection process of procedure (3), the drainage pipe is closed to form a closed space and apply internal pressure.

The procedure is clarified as follows:

“(1) Connect the water injection and drainage pipes to the test pipeline and inject water into the test pipeline through tap water. After the drainage pipe overflows for a period of time, close the outlet pipe and inlet valve to ensure that the test pipeline and booster cylinder are fully vented and filled with water. The water injection of procedure (1) mainly fills the internal space of the pipeline through water injection and discharges air.

(3) Inject water and pressurize the pipeline through pressurization device, measure the internal pressure load with high-precision pressure sensors, and transmit the sensor signal to the measurement control system to achieve feedback and record the internal pressure load value. The water injection of procedure (3) mainly involves injecting more water than the internal space of the pipeline into the pipeline to apply internal pressure.”

4) It should be noted that internal pressure is pushed by the pressure device in Fig. 5 (c) to move the connecting rod, thereby squeezing the water in the water tank into the pipeline and applying pressure. Therefore, there are two loading methods: one is force controlled loading, which controls the pressure value applied per unit time; Another type is displacement control, which controls the displacement of the connecting rod per unit time.

Force controlled loading is a commonly used loading method that can clarify the specific loading rate of the load. However, force controlled loading cannot always be used for pipeline damage tests, as at the moment of pipeline explosion, the enclosed space inside the pipeline is opened, and the pipeline is connected to the outside environment, causing the internal pressure of the pipeline to instantly decrease to 0. If force controlled loading is used at this time, it will cause the loading system to be unable to find the loading target value and cause the connecting rod to shake back and forth. Therefore, in this experiment, when the pipeline is approaching failure, it is converted to displacement control.

For the internal pressure failure of pipelines, the strain in the elastic stage increases

linearly with the internal pressure and is controlled by force; A small increase in internal pressure during the shaping stage can lead to significant deformation, which is controlled by displacement. Therefore, after yielding at multiple measurement points of pipeline in this experiment, displacement control was used instead.

Among them, "0.25MPa/min" represents an increase in internal pressure of 0.25MPa per minute, and "1.5mm/min" represents an increase in connecting rod of 1.5mm per minute, which is equivalent to injecting 0.3cm³ water into the pipeline every minute.

The loading method and loading rate were determined by referring to relevant experimental literature and analyzing multiple experimental data. **The relevant literature is as follows:**

- [1] Benjamin AC, Freire JLF, Vieira RD. Part 6: Analysis of pipeline containing interacting corrosion defects. EXP TECHNIQUES 2007;31(3):74-82.
- [2] Freire J, Vieira RD, Castro J, Benjamin AC. Part 3: Burst tests of pipeline with extensive longitudinal metal loss. EXP TECHNIQUES 2006;30(6):60-5.
- [3] Benjamin AC, Freire JLF, Vieira RD, Diniz JL, de Andrade EQ. Burst tests on pipeline containing interacting corrosion defects. ASME 2005 24th International Conference on Offshore Mechanics and Arctic Engineering. Halkidiki, Greece: American Society of Mechanical Engineers; 2005. p. 403-17.
- [4] Souza RD, Benjamin AC, Vieira RD, Freire J, Castro J. Part 4: Rupture tests of pipeline segments containing long real corrosion defects. EXP TECHNIQUES 2007;31:46-51.

In order to facilitate readers' understanding of the specific loading process, the content has been modified as follows:

“(4) Firstly, force controlled loading is used, with a loading rate of 0.25MPa/min. That is an increase in internal pressure of 0.25MPa per minute. As most measuring points enter the yield stage, switch to displacement control. The pressure loading rate was decreased and followed the volume injection rate of water until the pipeline fails due to burst, which was kept to a maximum of 0.3 cm³/min.”

8. Comment: (8) Page 13, Fig.5; 1) Add the explanation that the green marks are the strain gauge for hoop strain, and the red marks are those for axial strain. 2) Prepend “R” to numbers those represent the strain measurement points. (so as to correspond with the description in the text).

Author’s Response:

Thanks very much for the reviewer's suggestion! The revisions have been made in accordance with the reviewer's comments, as follows:

“

(Note:the green marks are the strain gauge for hoop strain, and the red marks are those for axial strain)

”

9. Comment: (9) Page 14, the fourth line in the section 4.4.3, “Considering the vertical placement and the difference of wall thickness”; 1) I cannot understand well the meaning of “vertical placement”. 2) Did the authors measure the actual wall thickness distribution? If the authors have information about the wall thickness, it is better to discuss the influence of wall thickness on the rupture failure with such evidence.

Author’s Response:

1) Vertical placement refers to the axial direction of the pipeline being perpendicular to the horizontal plane of the experimental equipment, as shown in Fig. 2.

For vertically placed pipelines, the inner walls of pipelines at different positions bear different hydrostatic pressures (generated by the gravity of the water inside the pipeline). The lower part of the experimental pipe section bears more hydrostatic pressure than the upper part, so the lower part of the experimental pipe section is more prone to damage.

However, the difference of hydrostatic pressure is very small compared to the internal pressure load applied by the pressurization device, so it will not affect the specific results of failure pressure .

Fig .2 Pipeline Location Diagram (Left View)

2) The authors have measured the actual wall thickness distribution. Fig. 3 shows the distribution of wall thickness along the axial and circumferential direction of the pipe line. It can be seen that the distribution is random in wall thickness, with significant changes along the circumferential direction and small changes in the axial direction. This is caused by the circumferential non-uniformity in the machining process.

Fig. 3 Geometry distribution of wall thickness

The wall thickness of the lower part of the pipeline in the circumferential direction of $252^{\circ} \sim 324^{\circ}$ is the smallest, with an average of 1.81 mm, and this position is the most prone to damage. The actual blasting position is in the middle and lower part of the 270° circumferential direction, which also verifies the wall thickness measurement results.

The influence of wall thickness on the rupture failure with information about the wall thickness have been added in the manuscript. The revised content is as follows:

“Fig. 8 shows the distribution of wall thickness along the axial and circumferential direction. It can be seen that the distribution is with significant changes along the circumferential direction and small changes in the axial direction. The wall thickness of the lower part of the pipeline in the circumferential direction of $252^{\circ} \sim 324^{\circ}$ is the smallest, with an average of 1.81 mm, and this position is the most prone to damage. The actual blasting position is in the middle and lower part of the 270° circumferential direction, which also verifies the wall thickness measurement results.

Fig. 8 Geometry distribution of wall thickness”

10. Comment: Page 14, Fig.6; please clarify the difference between those four photos.

Author’s Response:

There is no obvious difference between the four photos in Fig. 7. They are all used to describe the phenomenon of pipeline failure with internal pressure and axial force, but the shooting angles are different.

The four photos in Fig. 7 were reorganized and two typical schematic diagrams were retained. One is a global map of the failed pipeline, and the other is a local map of the failed location. The revised content is as follows:

“

(a) global diagram

(b) partial failure diagram

Fig. 7 Failure modes of intact pipelines”

11. Comment: (11) Page 15; to understand the discussion in this section, it is helpful to show where the failure location is in the strain measurement map (Fig.5(a)).

Author's Response:

The Fig.5(a) was revised while the failure location was marked. The revised content is as follows:

“

(a) schematic diagram(mm)

(Note:the green marks are the strain gauge for hoop strain, and the red marks are those for axial strain)

”

12. Comment: (12) Page 17, Table 3; add units for yield strength, tensile strength, and failure pressure.

Author's Response:

Thank you very much for the reviewer's suggestion. The revisions have been made as required. The revised content in Table 3 and **Appendix A** is as follows:

Table 3 Data of Burst tests

literature Sources	test No.	diameter (mm)	grade	D/t	axial stress (MPa)	yield strength (MPa)	tensile Strength (MPa)
	13	88.9	L80	14.5	348.79247	695.201	753.766
P.R. Paslay et al. (Paslay et al., 1998)	16	88.9	L80	18.0	586.66283	695.201	753.766
	17	88.9	L80	18.6	638.14491	695.201	753.766
	18	177.8	K55	26.7	454.31971	465.764	737.23
	19	196.85	Q125	18.1	884.73801	905.346	993.538
	22°C-1				450		
	22°C-2				450		
	22°C-3				460		
	22°C-4				550		
	22°C-5				540		
	22°C-6				650		
	22°C-7				655	969	1063
	22°C-8				665		
Lasebikan et al. (Lasebikan and Akisanya, 2014)*	22°C-9				750		
	22°C-10	8	125	0.25	751		
	22°C-11				740		
	22°C-12				750		
	90°C-1				400		
	90°C-2				410		
	90°C-3				410	881	948
	90°C-4				550		
	90°C-5				551		
	90°C-6				650		
	110°C-1				400	851	935
	110°C-2				400		

110°C-3	400		
110°C-4	550		
110°C-5	545		
110°C-6	550		
			
160°C-1	400		
160°C-2	400		
160°C-3	550	821	888
160°C-4	550		
160°C-5	650		

*Note: The data was obtained from the charts in the reference literature.

Appendix A Comparison of predicted values for different yield criteria (burst test data)

case	failure pressure (MPa)	predicted value of different yield criterias (MPa)				error analysis of different yield criterias			
		Tresca	ASSY	von Mises	TS	Tresca	ASSY	von Mises	TS
		(b=0)	(b=0.168)	(b=0.366)	(b=1)	(b=0)	(b=0.168)	(b=0.366)	(b=1)
test in the paper	11.69	11.38	11.97	12.57	13.94	-2.66%	2.42%	7.50%	19.25%
13	115.5	120	117.91	116.47	114.44	3.91%	2.10%	0.86%	-0.90%
16	64.1	39.32	55.16	60.18	72.23	-38.64%	-13.92%	-6.08%	12.72%
17	50.6	26.28	41.19	51.36	65.86	-48.11%	-18.67%	1.43%	30.06%
18	54.8	44.03	56.34	49.67	52.92	-19.71%	2.73%	-9.43%	-3.50%
19	63.4	25.45	48.24	63.81	85.96	-59.85%	-23.90%	0.66%	35.60%
22°C-1	63	66.44	67.23	68.01	69.83	5.46%	6.71%	7.96%	10.85%
22°C-2	60	66.44	67.23	68.01	69.83	10.73%	12.04%	13.36%	16.39%
22°C-3	58	66.44	67.13	67.82	69.42	14.55%	15.74%	16.93%	19.68%
22°C-4	65	64.13	64.71	65.10	65.67	-1.35%	-0.45%	0.16%	1.03%
22°C-5	63	65.38	65.64	65.82	66.08	3.77%	4.19%	4.48%	4.89%
22°C-6	63	66.44	65.29	64.15	61.50	5.46%	3.64%	1.82%	-2.38%
22°C-7	59	66.44	54.88	57.52	61.29	12.61%	-6.99%	-2.50%	3.88%
22°C-8	54	49.75	53.94	56.80	60.88	-7.87%	-0.11%	5.19%	12.73%
22°C-9	61.5	66.44	64.33	62.21	57.33	8.03%	4.59%	1.16%	-6.78%
22°C-10	59	66.44	64.32	62.19	57.29	12.61%	9.01%	5.41%	-2.90%
22°C-11	57	66.44	64.42	62.41	57.75	16.56%	13.02%	9.48%	1.32%
22°C-12	54.5	66.44	45.98	50.67	57.33	21.90%	-15.63%	-7.03%	5.20%

90°C-1	62.5	59.25	59.97	60.68	62.33	-5.20%	-4.06%	-2.91%	-0.27%
90°C-2	55	59.25	59.87	60.49	61.92	7.73%	8.85%	9.98%	12.58%
90°C-3	54	59.25	59.87	60.49	61.92	9.72%	10.87%	12.01%	14.66%
90°C-4	55	59.25	52.14	53.77	56.08	7.73%	-5.21%	-2.25%	1.97%
90°C-5	54.5	59.25	52.04	53.69	56.04	8.72%	-4.51%	-1.48%	2.83%
90°C-6	50.5	59.25	57.55	46.55	51.92	17.33%	13.96%	-7.82%	2.81%
110°C-1	65	58.44	59.09	63.31	61.25	-10.10%	-9.09%	-2.60%	-5.77%
110°C-2	61	58.44	59.09	59.74	61.25	-4.20%	-3.13%	-2.06%	0.41%
110°C-3	58.5	58.44	59.09	59.74	61.25	-0.11%	1.01%	2.12%	4.70%
110°C-4	56.5	58.44	57.64	56.84	55.00	3.43%	2.02%	0.61%	-2.65%
110°C-5	55	58.44	57.69	56.94	55.21	6.25%	4.89%	3.53%	0.38%
110°C-6	53.4	58.44	50.71	52.48	55.00	9.43%	-5.03%	-1.72%	3.00%
160°C-1	53	55.50	55.93	56.35	57.33	4.72%	5.52%	6.32%	8.18%
160°C-2	50.5	55.50	55.93	56.35	57.33	9.90%	10.74%	11.59%	13.53%
160°C-3	51.5	55.50	54.48	53.45	51.08	7.77%	5.78%	3.79%	-0.81%
160°C-4	48.5	42.25	45.58	47.85	51.08	-12.89%	-6.03%	-1.34%	5.33%
160°C-5	45.5	55.50	53.51	40.63	46.92	21.98%	17.60%	-10.70%	3.11%

Note: error = $(p_f - p_T) / p_T \times 100\%$. p_f is the predicted value, p_T is the failure pressure of test.”

13. Comment: (13) Page 18, Table 4; please provide the predicted pressure values by each prediction method. To notify again the actual failure pressure values in the tests in Table 4 is helpful.

Author’s Response:

The predicted pressure values by each prediction method have been provided and the actual failure pressure values in the tests have been notified . To verify the accuracy of the proposed method in this article, 29 additional experimental data were added. And **Appendix A** has been added to list the predicted and actual values..The revised content is as follows:

“

Table 4 The errors of new evaluation method (burst test)

index	error with different yield criterias			
	Tresca (b=0)	ASSY (b=0.168)	von Mises (b=0.366)	TS (b=1)
min	-0.11%	-0.11%	0.16%	-0.27%
max	-59.85%	-23.90%	16.93%	35.60%
average	12.60%	7.83%	5.26%	7.80%
Standard deviation	0.1278	0.0580	0.0429	0.0841

Note:error= $(P_f - P_T) / P_T \times 100\%$. P_f is the predicted value, P_T is the failure pressure of test. average= $\sum |error| / 35$

Appendix A Comparison of predicted values for different yield criteria (burst test data)

case	failure pressure (MPa)	predicted value of different yield criterias (MPa)				error analysis of different yield criterias			
		Tresca	ASSY	von Mises	TS	Tresca	ASSY	von Mises	TS
		(b=0)	(b=0.168)	(b=0.366)	(b=1)	(b=0)	(b=0.168)	(b=0.366)	(b=1)
test in the paper	11.69	11.38	11.97	12.57	13.94	-2.66%	2.42%	7.50%	19.25%
13	115.5	120	117.91	116.47	114.44	3.91%	2.10%	0.86%	-0.90%
16	64.1	39.32	55.16	60.18	72.23	-38.64%	-13.92%	-6.08%	12.72%
17	50.6	26.28	41.19	51.36	65.86	-48.11%	-18.67%	1.43%	30.06%
18	54.8	44.03	56.34	49.67	52.92	-19.71%	2.73%	-9.43%	-3.50%
19	63.4	25.45	48.24	63.81	85.96	-59.85%	-23.90%	0.66%	35.60%
22°C-1	63	66.44	67.23	68.01	69.83	5.46%	6.71%	7.96%	10.85%
22°C-2	60	66.44	67.23	68.01	69.83	10.73%	12.04%	13.36%	16.39%
22°C-3	58	66.44	67.13	67.82	69.42	14.55%	15.74%	16.93%	19.68%
22°C-4	65	64.13	64.71	65.10	65.67	-1.35%	-0.45%	0.16%	1.03%
22°C-5	63	65.38	65.64	65.82	66.08	3.77%	4.19%	4.48%	4.89%
22°C-6	63	66.44	65.29	64.15	61.50	5.46%	3.64%	1.82%	-2.38%
22°C-7	59	66.44	54.88	57.52	61.29	12.61%	-6.99%	-2.50%	3.88%
22°C-8	54	49.75	53.94	56.80	60.88	-7.87%	-0.11%	5.19%	12.73%
22°C-9	61.5	66.44	64.33	62.21	57.33	8.03%	4.59%	1.16%	-6.78%
22°C-10	59	66.44	64.32	62.19	57.29	12.61%	9.01%	5.41%	-2.90%
22°C-11	57	66.44	64.42	62.41	57.75	16.56%	13.02%	9.48%	1.32%
22°C-12	54.5	66.44	45.98	50.67	57.33	21.90%	-15.63%	-7.03%	5.20%

90°C-1	62.5	59.25	59.97	60.68	62.33	-5.20%	-4.06%	-2.91%	-0.27%
90°C-2	55	59.25	59.87	60.49	61.92	7.73%	8.85%	9.98%	12.58%
90°C-3	54	59.25	59.87	60.49	61.92	9.72%	10.87%	12.01%	14.66%
90°C-4	55	59.25	52.14	53.77	56.08	7.73%	-5.21%	-2.25%	1.97%
90°C-5	54.5	59.25	52.04	53.69	56.04	8.72%	-4.51%	-1.48%	2.83%
90°C-6	50.5	59.25	57.55	46.55	51.92	17.33%	13.96%	-7.82%	2.81%
110°C-1	65	58.44	59.09	63.31	61.25	-10.10%	-9.09%	-2.60%	-5.77%
110°C-2	61	58.44	59.09	59.74	61.25	-4.20%	-3.13%	-2.06%	0.41%
110°C-3	58.5	58.44	59.09	59.74	61.25	-0.11%	1.01%	2.12%	4.70%
110°C-4	56.5	58.44	57.64	56.84	55.00	3.43%	2.02%	0.61%	-2.65%
110°C-5	55	58.44	57.69	56.94	55.21	6.25%	4.89%	3.53%	0.38%
110°C-6	53.4	58.44	50.71	52.48	55.00	9.43%	-5.03%	-1.72%	3.00%
160°C-1	53	55.50	55.93	56.35	57.33	4.72%	5.52%	6.32%	8.18%
160°C-2	50.5	55.50	55.93	56.35	57.33	9.90%	10.74%	11.59%	13.53%
160°C-3	51.5	55.50	54.48	53.45	51.08	7.77%	5.78%	3.79%	-0.81%
160°C-4	48.5	42.25	45.58	47.85	51.08	-12.89%	-6.03%	-1.34%	5.33%
160°C-5	45.5	55.50	53.51	40.63	46.92	21.98%	17.60%	-10.70%	3.11%

Note: error = $(p_f - p_T) / p_T \times 100\%$. p_f is the predicted value, p_T is the failure pressure of test.

14. Comment: From Page 18, 5.2 The FE model; 1) provide the detail of FE analysis model. There are no description what kind of the FE code is used, what the FE model configuration is, what kind of elements are used, how the model is meshed, how the material properties are modeled, and so on. 2) the authors explained as $\frac{\sum(\sigma_L)}{\sigma_u}$ is taken as 0.25; however, in Table 5 and the related explanations, the authors describe that the value is set as 0.3. Which of the condition is correct? 3) in the note of Table 5, the authors describe “Failure pressure on the compression side or tensile side caused by bending moment”. I cannot understand well the meaning of this sentence. 4) Why the dimensions and the steel grade of pipes in the FE analysis are different from the experiments the authors conducted? Did the authors compare the experimental results with the FE analytical results? If not, how did the authors evaluate the reliability of the FE analysis model?

Author’s Response:

1) The details of the FE analysis model is provided, which is as follows:

“Establish a geometric model of the pipeline with ANSYS. Establish a finite element model of the pipeline using a three-dimensional solid unit C3D8R. Divide the pipeline into 4 layers of units in the thickness direction, 48 units in the circumferential direction, and 88 units in the axial direction. Set reference points at both ends of the pipeline, and use rigid beam constraints between the reference points and the pipeline end nodes. The analyzed model is as shown in Fig. 11.

Fig. 11. FE model for predicting the burst pressure

By applying the bending moment (M) and axial force (F) simultaneously to the reference node, the internal pressure on the inner surface nodes was increased gradually until the pipe experienced burst failure.

To solve the finite element model, the nonlinear arc-length method algorithm was employed. The simulation utilized a pipeline material exhibiting isotropic hardening plasticity. For the true stress-strain relationship, the well-established Ramberg-Osgood model was adopted.

The burst data of Table 4 was adopted to verify the correctness of the model and the results are listed in Table 5. It illustrates that the average error is 3.11%, which is within the acceptable level. The accuracy of the model has been verified.

Table 5 The errors of the FE model

steel grade	case	failure pressure in the test (MPa)	failure pressure with FE model (MPa)	error
Q235	test in the paper	11.69	11.70	0.09%
L80	13	115.48	111.67	-3.30%
L80	16	64.08	68.68	7.18%

L80	17	50.64	50.66	0.03%
K55	18	54.84	57.37	4.62%
Q125	19	63.39	61.20	-3.45%
	min			-3.45%
	max			7.18%
	average			3.11%
	Standard deviation			0.0427

Note: error = $(P_{FE} - P_T) / P_T \times 100\%$. P_{FE} is the predicted value, P_T is the failure pressure of test. average = $\sum|\text{error}|/6$

The size and strength data of X52 in the Yi Shuai and Xiao Zhang et. al. models were adopted for analysis (Shuai et al., 2022). The reasons for adopting the data are as follows:

(1) Due to the lack of pipeline data with medium strength in the burst test, X52 steel was adopted as a representative for analysis.

(2) At present, there is a lack of data on the large diameter to thickness ratio in the test (the diameter to thickness ratio of burst test in Table 4 are 14.5~73.0), therefore $D/t=96$ is adopted as a representative.

The specific finite element model is shown in Tab. 6.”

2) I am very sorry for the inconvenience caused by the author's mistake. $\frac{\sum(\sigma_L)}{\sigma_u} = 0.3$ is

correct, and the entire text has been corrected.

3) The pipeline involves compression and tension sides under bending moment, except for the neutral axis, as shown in Fig. 4. The axial force generated by the bending moment on the compression side is compressive stress, while the axial force generated by the bending moment on the stretching side is tensile stress. The internal pressure bearing capacity of the compression side and tension side are different, and the finite element model can analyze the ultimate orange bearing capacity as failure occurs at different positions.

Fig. 4 Outline configurations of pipeline with bending

According to the different stress models on the tensile and compressive sides, the evaluation model proposed in this paper can evaluate the failure pressure at different positions. Table 6 compares and analyzes the failure pressure at the tension and compression sides for the following two purposes:

(1) Compare the internal pressure bearing capacity of the tensile and compressive sides with the same case, and determine the most likely failure location of the pipeline with bending moment action

(2) The axial force value caused by bending moment on the compression and tension sides is the same, but in opposite directions. Therefore, comparing the internal pressure bearing capacity of the tension and compression sides with the same case can clarify the differences in the influence of tensile and compressive stresses on failure pressure.

For example, by comparing the cases "C-T0.3-B0.05" and "T-T0.3-B0.05", as well as "C-T0.3-B0.05" and "T-T0.3-B0.3", it can be seen that under the axial force being tensile stress, the bearing capacity of the compression side is greater than that of the tension side, and the pipeline is more prone to failure on the tension side; Comparing the cases "C-C0.3-B0.05" and "T-C0.3-B0.05", as well as "C-C0.3-B0.3" and "T-C0.3-B0.3", it can be seen that under

the axial force being compressive stress, the bearing capacity of the compression side is smaller than that of the tension side, and the pipeline is more prone to failure on the compression side.

The influence of bending moment on the bearing capacity of pipelines will change due to different axial force properties.

Table 6 Results of finite element model

test No.	diameter (mm)	grade	yield strength (MPa)	tensile Strength (MPa)	t	axial Stress (MPa)	bend moment (N • mm)	failure pressure (MPa)
C-C0.3-B0.3						$-0.3\sigma_u$	$0.3M_0$	9.11
C-C0.3-B0.05						$-0.3\sigma_u$	$0.05M_0$	9.94
C-T0.3-B0.3						$0.3\sigma_u$	$0.3M_0$	13.20
C-T0.3-B0.05	914.4	X52	413	545	9.525	$0.3\sigma_u$	$0.05M_0$	14.39
T-C0.3-B0.3						$-0.3\sigma_u$	$0.3M_0$	12.13
T-C0.3-B0.05						$-0.3\sigma_u$	$0.05M_0$	10.41
T-T0.3-B0.3						$0.3\sigma_u$	$0.3M_0$	12.13
T-T0.3-B0.05						$0.3\sigma_u$	$0.05M_0$	10.41

4) The finite element model and failure pressure solution method used in this article have been verified through full-scale pipeline experiments, and verification content has been added to the revised manuscript. The reason why the experimental data in this article was not used is mainly because the finite element model is mainly used to compensate for the limitations of the experiment and increase the comprehensiveness of the validation data. Therefore, based on existing research literature on bearing capacity of pipeline, the new data was adopted.

The size and strength data of X52 in the Yi Shuai and Xiao Zhang et. al. models were adopted for analysis (Shuai et al., 2022). The reasons for adopting the data are as follows:

(1) Due to the lack of pipeline data with medium strength in the burst test, X52 steel was

adopted as a representative for analysis.

(2)At present, there is a lack of data on the large diameter to thickness ratio in the test (the diameter to thickness ratio of burst test in Table 4 are 14.5~73.0), therefore $D/t=96$ is adopted as a representative.

Reference

Shuai, Y., Zhang, X., Feng, C., Han, J., Cheng, Y.F., 2022. A Novel Model for Prediction of Burst Capacity of Corroded Pipelines Subjected to Combined Loads of Bending Moment and Axial Compression. *Int J. Pres Ves Pip.* 196, 104621.

Date: Dec. 26, 2023

To: Nature Communications

From: SUN Ming-ming, FANG Hong-yuan*, WANG Nian-nian, DU Xue-ming, Zhai Ke-Jie

Re: Manuscript No.: NCOMMS-23-43457

Title: Research on the Limit State Equation and Failure Pressure Prediction Model of Pipeline with Complex Loading

Thank you for taking the time to review our submission. Your Comments are invaluable and significant to improve our paper. We have carried out all the corrections as suggested. The Comments for this paper and the response to these comments are as following. The revised content in the manuscript has been highlighted in yellow.

Reviewer #2

1. Comment: What are the main innovations of this manuscript? Many failure pressure evaluation models are proposed every year. What is special about the evaluation model in this manuscript?

Author's Response:

At present, there are many evaluation models for the bearing capacity of pipelines with individual internal pressure loads, but there are few models for evaluating the failure pressure of pipelines under complex loads, and the accuracy needs to be verified. The failure pressure assessment models for pipelines with multiple loads are mainly divided into the following four types:

1) Evaluation methods provided by specifications or reports

□ The widely used standard DNV-RP-F101 (DNV-RP-F101, 2015) provides a burst model for corroded pipelines, considering the effect of external loads by modifying the failure pressure model through an external compressive longitudinal stress factor H_1 .

The burst pressure p_f , is obtained by:

$$p_f = \frac{2t\sigma_b}{D-t} \left(\frac{1-(d/t)}{1-(d/t)/Q} \right) H_1 \quad (1)$$

$$H_1 = \frac{1 + \frac{\sigma_L}{\sigma_b A_r}}{1 - \frac{1}{2A_r} \frac{1-(d/t)}{1-\frac{(d/t)}{Q}}} \quad (2)$$

where D is the diameter; d and w are the depth and width of corrosion defect; t is the wall thickness; Q is the correction factor for corrosion length; A_r is the circumferential area reduction factor: $A_r = 1 - \frac{dw}{\pi Dt}$; σ_L is axial stress; σ_b is tensile strength. This model is suitable

especially as the corrosion defect is withstand compressive stress. **However, as additional longitudinal stresses in the corrosion region $\sigma_L < 0.25\sigma_b$, the prediction accuracy remarkably decreased.**

□ Chauhan (Chauhan and Swankie, 2015) and Liu et al. (Liu et al., 2009) proposed interaction diagrams to evaluate the ultimate internal pressure of corroded pipelines under a single bending load or axial force. **However, this method cannot be applied on pipelines that are subject to a simultaneous loading of both bending moment and axial force.**

2) The evaluation methods provided by using finite element data fitting

□ A prediction model was proposed for determination of burst capacity of corroded pipelines under an axial stress by Pengcheng et al. (Pengcheng et al., 2019). **However, the model was not applicable to a non-uniform axial stress caused by bending moment since it was developed based on the assumption that the cross-section of the pipe was under the same axial stress.**

□ Zhou et al. (Zhou et al., 2018) investigated the effect of a longitudinal strain caused by bending load on the burst capacity of a corroded pipeline by full-scale burst testing, and found that the longitudinal strain at defective area reduced the burst pressure. Based on FE analysis, a strain-based burst pressure prediction formula for pipelines containing a corrosion defect was developed. **However, the model was applicable for small defects only.**

□ A three-dimensional (3D) nonlinear finite element (FE) model was developed by Yi Shuai et al. (Shuai et al., 2022) to investigate the effect of bending moment and axial force on the burst capacity of corroded pipelines. Then based on a series of FE cases, a new burst prediction model for corroded pipelines subjected to the combination loads of bending load and axial compressive force was fitted and developed.

However, the finite element model and prediction formula have not been verified by intact pipeline failure experiments with both axial force and bending moment. And the evaluation model was fitted based on numerical results and did not theoretically derive an evaluation model for the internal pressure bearing capacity of pipelines under complex loads

□ Considering the axial compression and closing bending moment, failure loci of combined bending moments and internal pressures are developed for different axial forces by Mondal and Dhar (Mondal and Dhar, 2019). The developed failure loci can be used for assessing the burst pressure of corroded pipelines subjected to axial forces and bending moments. **For cases where the failure loci is not presented, the applicability of this method is reduced.**

3) Evaluation method with computational procedures

① Conventional work-hardening plasticity theory was used to establish a computer program for the prediction of the burst pressure in the presence of an axial load by Paslay et al. (Paslay et al., 1998). **However, the program was run extensively with the stress–strain curve, and does not provide a simple failure pressure prediction formula. This program is not applicable to the evaluation of pipeline failure pressure under bending moment**

action.

□ A detailed analysis of the rupture and necking of tubulars is provided by Klever (Klever, 2006) and Stewart et al. (Stewart et al., 1994). The minimum internal pressure necessary for rupture is given, assuming von Mises yield criterion. **However, the axial effective stress was evaluated using this method and the influence of external axial stress magnitude and direction on the failure pressure was not directly given. And the axial effective stress changes with the magnitude of internal pressure, so it cannot be directly calculated for failure pressure.**

$$p_f = \frac{2}{(\sqrt{3})^{n+1}} \frac{2t\sigma_b}{D-t} \sqrt{1 - \frac{4^{1-n} - 1}{3^{1-n}} \left(\frac{\sigma_{eff}}{\sigma_b} \right)} \quad (3)$$

Where σ_{eff} is the effective stress in axial direction.

4) Evaluation methods of semi-empirical equation

Chen et al. (Chen et al., 2021) first proposed semi-empirical failure pressure equation of corroded pipelines subjected to internal pressure and axial tension, which is an improvement on the theory of evaluating the bearing capacity of pipelines with complex loads. **However, this method does not consider bending moment loads, and its accuracy remains to be confirmed for pipelines with other types of defects.**

Compared to existing evaluation methods, the evaluation method proposed in this article has the **following advantages**:

1) The existing evaluation methods are mainly divided into two types: one is based on the improvement of the bearing capacity evaluation model with the single load (internal pressure), and the other is based on the fitting of numerical calculation results. **The existing evaluation methods do not provide a limit state equation for pipelines from the perspective of mechanical mechanisms**, which is not conducive to theoretically analyzing the impact of load size and type on the internal pressure bearing capacity of pipelines.

2) The existing evaluation models do not consider the impact of differences in yield

criteria on the evaluation results. **The evaluation model proposed in the article is an integration of multiple yield criteria, and can be transformed into an evaluation model based on a single yield criterion by changing parameter b .** Therefore, this model can compare and analyze the evaluation results of different yield criteria, and thus clarify the optimal yield criteria applicable to different load modes.

3) Different application methods of external loads can affect the evaluation results of failure pressure, such as axial pressure reducing the internal pressure bearing capacity of pipelines, while axial tension increasing the internal pressure bearing capacity. **The method proposed in the article proposes a corresponding evaluation model based on the differences in the distribution of principal stresses (which is shown in Table 1 in the manuscript.).** Compared to the singularity of existing evaluation methods, it has a wider applicability.

4) **The analysis method and theory proposed in the article have broader application prospects, and can be converted into bearing capacity evaluation of different failure modes based on changes in equivalent stress σ_{UE} (as shown in Eq. (4)).** For example, for internal pressure or axial tensile failure, the ultimate tensile strength can be selected as equivalent stress σ_{UE} for evaluation; For buckling failure, equivalent stress σ_{UE} can be adopted as evaluate buckling stress; For plastic failure, the equivalent stress σ_{UE} can be adopted as the yield stress for evaluation.

$$\sigma_{UE} = \begin{cases} \sigma_1 - \frac{1}{1+b}(b\sigma_2 + \sigma_3), & \sigma_2 \leq \frac{\sigma_1 + \sigma_3}{2} \\ \frac{1}{1+b}(\sigma_1 + b\sigma_2) - \sigma_3, & \sigma_2 > \frac{\sigma_1 + \sigma_3}{2} \end{cases} \quad (4)$$

Reference

- DNV-RP-F101, 2015. Recommended Practice DNV-RP-F101 Corroded Pipelines. Det-Norske-Veritas, Norway.
- Chauhan, V., Swankie, T., 2015. Guidance for Assessing the Remaining Strength of Corroded Pipelines.

- Pipeline and Hazardous Materials Safety Administration, Washington, DC, USA: US Department of Transportation.
- Liu, J., Chauhan, V., Ng, P., Wheat, S., Hughes, C., 2009. Remaining Strength of Corroded Pipe Under Secondary (Biaxial) Loading. GL Industrial Services UK Ltd.
- Pengcheng, Z., Jian, S., Yu, T., Kui, X.U., 2019. Impact of Axial Stress On Ultimate Internal Pressure of Corroded Pipelines. *China Safety Science Journal*. 29(3), 70-75.
- Zhou, H., Wang, Y., Stephens, M., Bergman, J., Nanney, S., 2018. Burst Pressure of Pipelines with Corrosion Anomalies Under High Longitudinal Strains. 12th International Pipeline Conference, Calgary, Alberta, Canada. pp. V2T-V6T.
- Shuai, Y., Zhang, X., Feng, C., Han, J., Cheng, Y.F., 2022. A Novel Model for Prediction of Burst Capacity of Corroded Pipelines Subjected to Combined Loads of Bending Moment and Axial Compression. *Int J. Pres Ves Pip*. 196, 104621.
- Mondal, B.C., Dhar, A.S., 2019. Burst Pressure of Corroded Pipelines Considering Combined Axial Forces and Bending Moments. *Eng Struct*. 186, 43-51.
- Paslay, P.R., Cernocky, E.P., Wink, R., 1998. Burst Pressure Prediction of Thin-Walled, Ductile Tubulars Subjected to Axial Load. SPE Applied Technology Workshop on Risk Based Design of Well Casing and Tubing, Woodlands, Texas. pp. 48327.
- Klever, F.J., 2006. Formulas for Rupture, Necking, and Wrinkling of Octg Under Combined Loads. SPE Annual Technical Conference and Exhibition, San Antonio, Texas, U.S.A. pp. 102585.
- Stewart, G., Klever, F.J., Ritchie, D., 1994. An Analytical Model to Predict the Burst Capacity of Pipelines. 13th International Offshore Mechanics and Arctic Engineering Conference, Houston, Texas. pp. 177-188.
- Chen, Z., Wang, W., Yang, H., Yan, S., Jin, Z., 2021. On the Effect of Long Corrosion Defect and Axial Tension On the Burst Pressure of Subsea Pipelines. *Appl Ocean Res*. 111, 102637.

2. Comment: In abstract, “The current evaluation method for failure pressure of steel pipelines does not consider the impact of other loads besides internal pressure.” As far as I know, many scholars have studied the failure pressure of pipelines under complex loads. Such as Shuai Y, Zhang X, Feng C, et al. A novel model for prediction of burst capacity of corroded pipelines subjected to combined loads of bending moment and axial compression[J]. International Journal of Pressure Vessels and Piping, 2022, 196: 104621. Chen Z F, Wang W, Yang H, et al. On the effect of long corrosion defect and axial tension on the burst pressure of subsea pipelines[J]. Applied Ocean Research, 2021, 111: 102637. Personally, I don't think that's accurate enough.

Author's Response:

I fully agree with the reviewer's viewpoint and apologize for the author's mistake. At present, the limit state equation of pipelines with different yield criteria is needed, which is the basis for evaluating the ultimate bearing capacity of pipelines. Moreover, there is relatively little research on the limit state equations of pipelines with different yield criteria, and most researchers adopt data fitting, semi-empirical equation or single yield criterion for analysis.

The “Introduction” section discusses the importance of studying “theoretical derivation of the limit state equation for pipelines with complex loads’ and “the differences in yield criteria”. Meanwhile, references "Shuai Y, Zhang X, Feng C, et al. A novel model for prediction of burst capacity of corroded pipelines subjected to combined loads of bending moment and axial compression[J]. International Journal of Pressure Vessels and Piping, 2022, 196: 104621" and "Chen Z F, Wang W, Yang H, et al. On the effect of long corrosion defect and axial tension on the burst pressure of subsea pipelines[J]. Applied Ocean Research, 2021, 111: 102637." are also cited. The abstract and introduction are revised as follows:

Abstract: “The limit state equation of steel pipes with different yield criteria is required, which is the basis for evaluating the ultimate bearing capacity of pipelines.”

Introduction: “Chen et al. (Chen et al., 2021) first proposed semi-empirical failure pressure equation of corroded pipelines subjected to internal pressure and axial tension,

which is an improvement on the theory of evaluating the bearing capacity of pipelines with complex loads. However, this method does not consider bending moment loads, and its accuracy remains to be confirmed for pipelines with other types of defects.”

3. Comment: There is considerable debate as to which is the more accurate criterion for the strength of metallic materials under complex loads. Which strength criterion is accurate may require extensive experimentation to give a suitable conclusion. There is less experimental data in this paper and it is recommended to find some experimental data from the literature to fully validate the conclusion.

Author’s Response:

I fully agree with the reviewer's suggestion. The author of the manuscript conducted a statistical analysis of the experimental data on pipeline with complex loads that have been publicly published. The current failure experiments of pipelines with complex loads mainly focus on "buckling failure under axial force and internal pressure", such as: (Shuai et al., 2021); “rupture, necking, wrinkling or other failure mode of pipelines with complex loads”, such as: (Klever, 2006), (Smith et al., 1998), (Roy et al., 1997), (Grigory and Smith, 1996); "bending failure under complex loads", such as: (Phan et al., 2021), (Chegeni et al., 2019); "failure pressure of corroded pipeline with complex loads", such as: (Wang et al., 1998), (Mondal and Dhar, 2019), (Smith and Waldhart, 2000), (Roberts and Pick, 1998), (Bhardwaj et al., 2022), (Chauhan and Swankie, 2015), (Smith and Grigory, 1996), (Benjamin, 2008), (Liu et al., 2009); "internal pressure failure of defect-free pipeline under complex loads", such as: (Lasebikan and Akisanya, 2014), (Paslay et al., 1998).

The manuscript mainly focuses on the study of internal pressure failure of intact pipelines under complex loads. Currently, only the experimental data provided by Paslay et al. which has been publicly published, is used in the text. At the same time, in order to make up for the shortcomings of existing experimental data, the manuscript author conducted experiments on low strength steel.

In order to increase the amount of data, Lasebikan and Akisanya's data has been added to the article for analysis. The number of experimental data has **increased from 6 to 35**, which is shown as follows:

“

5.1 Burst tests

The full-scale burst tests data with axial force and internal pressure performed by P.R. Paslay et al. (Paslay et al., 1998) and Lasebikan et al. (Lasebikan and Akisanya, 2014) were adopted and listed in Tab. 3. To indicate the superiority of the limit state equation in estimating failure pressure, the error distribution of 35 burst tests are compared in Tab. 4. The detailed predicted values and error data are shown in Appendix A. As shown in Tab. 4, the different error parameters are compared for the new evaluation method with different yield criterions.

Table 3 Data of Burst tests

literature Sources	test No.	diameter (mm)	grade	D/t	axial stress (MPa)	yield strength (MPa)	tensile Strength (MPa)
	13	88.9	L80	14.5	348.79247	695.201	753.766
P.R. Paslay et al. (Paslay et al., 1998)	16	88.9	L80	18.0	586.66283	695.201	753.766
	17	88.9	L80	18.6	638.14491	695.201	753.766
	18	177.8	K55	26.7	454.31971	465.764	737.23
	19	196.85	Q125	18.1	884.73801	905.346	993.538
	22°C-1				450		
	22°C-2				450		
	22°C-3				460		
Lasebikan et al. (Lasebikan and Akisanya, 2014)*	22°C-4				550		
	22°C-5	8	125	0.25	540	969	1063
	22°C-6				650		
	22°C-7				655		
	22°C-8				665		
	22°C-9				750		
	22°C-10				751		

22°C-11	740		
22°C-12	750		
90°C-1	400		
90°C-2	410		
90°C-3	410	881	948
90°C-4	550		
90°C-5	551		
90°C-6	650		
110°C-1	400		
110°C-2	400		
110°C-3	400	851	935
110°C-4	550		
110°C-5	545		
110°C-6	550		
160°C-1	400		
160°C-2	400		
160°C-3	550	821	888
160°C-4	550		
160°C-5	650		

*Note: The data was obtained from the charts in the reference literature.

From Tab. 4, it can be seen that among the four conventional yield criteria, von Mises criterion has the best adaptability and stability, with an average error of only 5.26%, which is smaller than the predicted error of other yield criteria. And the standard deviation is only 0.0429, which is the minimum value among the four yield criteria. The Tresca yield criterion has the worst adaptability, with not only the largest average error, but also a large dispersion of prediction results. It can be seen that for the analysis of pipeline burst failure with complex loads, the influence of the second principal stress should be considered. The burst failure of pipelines with complex loads is the comprehensive result of the action of the first principal

stress and second principal stress. The average error and standard deviation of the yield criteria ASSY and TS are not significantly different, and the applicability of ASSY is better (Standard deviation of ASSY is 0.0580, which is smaller than TS). The ASSY prediction results are conservative, while the TS prediction results are dangerous. The factor b in the von Mises criterion that determines the impact extent of the second principal stress on the failure stress is between the ASSY and TS yield criteria. The von Mises criterion is more in line with the characteristics of pipeline blasting failure under complex loads, so its prediction results are more accurate.

Table 4 The errors of new evaluation method (burst test)

index	error with different yield criterias			
	Tresca (b=0)	ASSY (b=0.168)	von Mises (b=0.366)	TS (b=1)
min	-0.11%	-0.11%	0.16%	-0.27%
max	-59.85%	-23.90%	16.93%	35.60%
average	12.60%	7.83%	5.26%	7.80%
Standard deviation	0.1278	0.0580	0.0429	0.0841

Note: error = $(P_f - P_T) / P_T \times 100\%$. P_f is the predicted value, P_T is the failure pressure of test. average = $\sum |error| / 35$

Appendix A Comparison of predicted values for different yield criteria (burst test data)

case	failure pressure (MPa)	predicted value of different yield criterias (MPa)				error analysis of different yield criterias			
		Tresca	ASSY	von Mises	TS	Tresca	ASSY	von Mises	TS
		(b=0)	(b=0.168)	(b=0.366)	(b=1)	(b=0)	(b=0.168)	(b=0.366)	(b=1)
test in the paper	11.69	11.38	11.97	12.57	13.94	-2.66%	2.42%	7.50%	19.25%
13	115.5	120	117.91	116.47	114.44	3.91%	2.10%	0.86%	-0.90%
16	64.1	39.32	55.16	60.18	72.23	-38.64%	-13.92%	-6.08%	12.72%
17	50.6	26.28	41.19	51.36	65.86	-48.11%	-18.67%	1.43%	30.06%
18	54.8	44.03	56.34	49.67	52.92	-19.71%	2.73%	-9.43%	-3.50%
19	63.4	25.45	48.24	63.81	85.96	-59.85%	-23.90%	0.66%	35.60%
22°C-1	63	66.44	67.23	68.01	69.83	5.46%	6.71%	7.96%	10.85%
22°C-2	60	66.44	67.23	68.01	69.83	10.73%	12.04%	13.36%	16.39%
22°C-3	58	66.44	67.13	67.82	69.42	14.55%	15.74%	16.93%	19.68%
22°C-4	65	64.13	64.71	65.10	65.67	-1.35%	-0.45%	0.16%	1.03%
22°C-5	63	65.38	65.64	65.82	66.08	3.77%	4.19%	4.48%	4.89%
22°C-6	63	66.44	65.29	64.15	61.50	5.46%	3.64%	1.82%	-2.38%
22°C-7	59	66.44	54.88	57.52	61.29	12.61%	-6.99%	-2.50%	3.88%
22°C-8	54	49.75	53.94	56.80	60.88	-7.87%	-0.11%	5.19%	12.73%
22°C-9	61.5	66.44	64.33	62.21	57.33	8.03%	4.59%	1.16%	-6.78%
22°C-10	59	66.44	64.32	62.19	57.29	12.61%	9.01%	5.41%	-2.90%
22°C-11	57	66.44	64.42	62.41	57.75	16.56%	13.02%	9.48%	1.32%
22°C-12	54.5	66.44	45.98	50.67	57.33	21.90%	-15.63%	-7.03%	5.20%

90°C-1	62.5	59.25	59.97	60.68	62.33	-5.20%	-4.06%	-2.91%	-0.27%
90°C-2	55	59.25	59.87	60.49	61.92	7.73%	8.85%	9.98%	12.58%
90°C-3	54	59.25	59.87	60.49	61.92	9.72%	10.87%	12.01%	14.66%
90°C-4	55	59.25	52.14	53.77	56.08	7.73%	-5.21%	-2.25%	1.97%
90°C-5	54.5	59.25	52.04	53.69	56.04	8.72%	-4.51%	-1.48%	2.83%
90°C-6	50.5	59.25	57.55	46.55	51.92	17.33%	13.96%	-7.82%	2.81%
110°C-1	65	58.44	59.09	63.31	61.25	-10.10%	-9.09%	-2.60%	-5.77%
110°C-2	61	58.44	59.09	59.74	61.25	-4.20%	-3.13%	-2.06%	0.41%
110°C-3	58.5	58.44	59.09	59.74	61.25	-0.11%	1.01%	2.12%	4.70%
110°C-4	56.5	58.44	57.64	56.84	55.00	3.43%	2.02%	0.61%	-2.65%
110°C-5	55	58.44	57.69	56.94	55.21	6.25%	4.89%	3.53%	0.38%
110°C-6	53.4	58.44	50.71	52.48	55.00	9.43%	-5.03%	-1.72%	3.00%
160°C-1	53	55.50	55.93	56.35	57.33	4.72%	5.52%	6.32%	8.18%
160°C-2	50.5	55.50	55.93	56.35	57.33	9.90%	10.74%	11.59%	13.53%
160°C-3	51.5	55.50	54.48	53.45	51.08	7.77%	5.78%	3.79%	-0.81%
160°C-4	48.5	42.25	45.58	47.85	51.08	-12.89%	-6.03%	-1.34%	5.33%
160°C-5	45.5	55.50	53.51	40.63	46.92	21.98%	17.60%	-10.70%	3.11%

Note: error = $(p_f - p_T) / p_T \times 100\%$. p_f is the predicted value, p_T is the failure pressure of test.

Reference

- Wang, W., Smith, M.Q., Popelar, C.H., Maple, J.A., 1998. A New Rupture Prediction Model for Corroded Pipelines Under Combined Loadings. The 2nd International Pipeline Conference, Calgary, Alberta, Canada. pp. 563-572.
- Mondal, B.C., Dhar, A.S., 2019. Burst Pressure of Corroded Pipelines Considering Combined Axial Forces and Bending Moments. Eng Struct. 186, 43-51.
- Smith, M.Q., Waldhart, C.J., 2000. Combined Loading Tests of Large Diameter Corroded Pipelines. The 3rd International Pipeline Conference, Calgary, Alberta, Canada. pp. V2T-V6T.
- Roy, S., Grigory, S., Smith, M., Kanninen, M.F., Anderson, M., 1997. Numerical Simulations of Full-Scale Corroded Pipe Tests with Combined Loading. Journal of Pressure Vessel Technology. 119(4), 457-466.
- Grigory, S.C., Smith, M.Q., 1996. Residual Strength of 48-Inch Diameter Corroded Pipe Determined by Full Scale Combined Loading Experiments. 1st International Pipeline Conference, Calgary, Alberta, Canada. pp. 377-386.
- Roberts, K.A., Pick, R.J., 1998. Correction for Longitudinal Stress in the Assessment of Corroded Line Pipe. The 2nd International Pipeline Conference, Calgary, Alberta, Canada. pp. 553-561.
- Bhardwaj, U., Teixeira, A.P., Soares, C.G., 2022. Failure Assessment of Corroded Ultra-High Strength Pipelines Under Combined Axial Tensile Loads and Internal Pressure. Ocean Eng. 257, 111438.
- Chauhan, V., Swankie, T., 2015. Guidance for Assessing the Remaining Strength of Corroded Pipelines. Pipeline and Hazardous Materials Safety Administration, Washington, DC, USA: US Department of Transportation.
- Smith, M.Q., Grigory, S.C., 1996. New Procedures for the Residual Strength Assessment of Corroded Pipe Subjected to Combined Loads. The 1st International Pipeline Conference, Calgary, Alberta, Canada. pp. 387-400.
- Benjamin, A.C., 2008. Prediction of the Failure Pressure of Corroded Pipelines Subjected to a Longitudinal Compressive Force Superimposed to the Pressure Loading. 7th International Pipeline Conference, Calgary, Alberta, Canada. pp. 179-189.

- Liu, J., Chauhan, V., Ng, P., Wheat, S., Hughes, C., 2009. Remaining Strength of Corroded Pipe Under Secondary (Biaxial) Loading. GL Industrial Services UK Ltd, Washington, DC, USA: US Department of Transportation.
- Shuai, Y., Wang, X., Feng, C., Zhu, Y., Wang, C., Sun, T., Han, J., Cheng, Y.F., 2021. A Novel Strain-Based Assessment Method of Compressive Buckling of X80 Corroded Pipelines Subjected to Bending Moment Load. *Thin Wall Struct.* 167, 108172.
- Klever, F.J., 2006. Formulas for Rupture, Necking, and Wrinkling of Octg Under Combined Loads. SPE Annual Technical Conference and Exhibition, San Antonio. Texas, U.S.A. pp. 102585.
- Smith, M.Q., Nicolella, D.P., Waldhart, C.J., 1998. Full-Scale Wrinkling Tests and Analyses of Large Diameter Corroded Pipes. The 2nd International Pipeline Conference, Calgary, Alberta, Canada. pp. 543-551.
- Phan, H.C., Le, T., Bui, N.D., Duong, H.T., Pham, T.D., 2021. An Empirical Model for Bending Capacity of Defected Pipe Combined with Axial Load. *Int J. Pres Ves Pip.* 191, 104368.
- Chegeni, B., Jayasuriya, S., Das, S., 2019. Effect of Corrosion On Thin-Walled Pipes Under Combined Internal Pressure and Bending. *Thin Wall Struct.* 143, 106218.
- Lasebikan, B.A., Akisanya, A.R., 2014. Burst Pressure of Super Duplex Stainless Steel Pipes Subject to Combined Axial Tension, Internal Pressure and Elevated Temperature. *Int J. Pres Ves Pip.* 119, 62-68.
- Paslay, P.R., Cernocky, E.P., Wink, R., 1998. Burst Pressure Prediction of Thin-Walled, Ductile Tubulars Subjected to Axial Load. SPE Applied Technology Workshop on Risk Based Design of Well Casing and Tubing, Woodlands, Texas. pp. 48327.

4. Comment: How are Equation 15 and Equation 17 obtained from Equation 6? Please explain.

Author's Response:

The process of obtaining formula (15) is as follows:

Hoop stress σ_h and axial stress $(\sigma_L)_p$ can be obtained as Eq.(6) and (7). Radial stress σ_r can be ignored relative to other stresses.

$$\sigma_h = p \frac{D_e}{2t} \quad (1)$$

$$(\sigma_L)_p = \frac{\sigma_h}{2} \quad (2)$$

The UYC criteria is shown as Eq.(3)

$$\sigma_{UE} = \begin{cases} \sigma_1 - \frac{1}{1+b} (b\sigma_2 + \sigma_3) , & \sigma_2 \leq \frac{\sigma_1 + \sigma_3}{2} \\ \frac{1}{1+b} (\sigma_1 + b\sigma_2) - \sigma_3 , & \sigma_2 > \frac{\sigma_1 + \sigma_3}{2} \end{cases} \quad (3)$$

where σ_{UE} is the UYC equivalent stress.

1) As the additional loadings is positive (tensile) and $\sigma_1 = \sigma_L$, $\sigma_2 = \sigma_h$ and $\sigma_3 = 0$.

(1) If $\sigma_2 \leq \frac{\sigma_1 + \sigma_3}{2}$, according to Eq.(3):

$$\sigma_{UE} = \sigma_1 - \frac{1}{1+b} (b\sigma_2 + \sigma_3) = \sigma_u \quad (4)$$

$$\sigma_1 = (\sigma_L)_p + \Sigma (\sigma_L) \quad (5)$$

$$\sigma_u = p_0 \frac{D_e}{2t} \quad (6)$$

Where p_0 is the internal pressure bearing capacity of the intact pipeline.

By substituting Eqs. (1) and (5) into Eq. (4), it can be obtained that:

$$(\sigma_L)_p + \Sigma (\sigma_L) - \frac{b}{1+b} * p \frac{D_e}{2t} = \sigma_u \quad (7)$$

$$\frac{(\sigma_L)_p}{\sigma_u} + \frac{\Sigma (\sigma_L)}{\sigma_u} - \frac{b}{1+b} \frac{p \frac{D_e}{2t}}{\sigma_u} = 1 \quad (8)$$

Substitute Eq. (2) and (6) into Eq. (8), it can be obtained that:

$$\frac{1}{2} * p \frac{D_e}{2t} + \frac{\Sigma (\sigma_L)}{\sigma_u} - \frac{b}{1+b} \frac{p \frac{D_e}{2t}}{p_0 \frac{D_e}{2t}} = 1 \quad (9)$$

$$\frac{1}{2} * \frac{p}{p_0} + \frac{\Sigma (\sigma_L)}{\sigma_u} - \frac{b}{1+b} \frac{p}{p_0} = 1 \quad (10)$$

$$\left(\frac{1}{2} - \frac{b}{1+b}\right) \frac{p}{p_0} + \frac{\Sigma (\sigma_L)}{\sigma_u} = 1 \quad (11)$$

This is the derivation process of Eq. (15) in the old version manuscript (Eq. (11) in this document). In order to prevent readers from misunderstanding the theoretical derivation process, the manuscript has been supplemented with the followings:

“

By substituting Eqs. (13) and (6) into Eq. (12), it can be obtained that:

$$(\sigma_L)_p + \Sigma (\sigma_L) - \frac{b}{1+b} * p \frac{D_e}{2t} = \sigma_u \quad (15)$$

$$\frac{(\sigma_L)_p}{\sigma_u} + \frac{\Sigma (\sigma_L)}{\sigma_u} - \frac{b}{1+b} \frac{p \frac{D_e}{2t}}{\sigma_u} = 1 \quad (16)$$

Substitute Eqs. (6), (7) and (14) into Eq. (16), it can be obtained that:

$$\frac{\frac{1}{2} * p \frac{D_e}{2t}}{p_0 \frac{D_e}{2t}} + \frac{\Sigma (\sigma_L)}{\sigma_u} - \frac{b}{1+b} \frac{p \frac{D_e}{2t}}{p_0 \frac{D_e}{2t}} = 1 \quad (17)$$

The limit state equation is defined as:

$$\left(\frac{1}{2} - \frac{b}{1+b}\right) \frac{p}{p_0} + \frac{\Sigma (\sigma_L)}{\sigma_u} = 1 \quad (18)$$

”

The process of obtaining formula (17) is as follows:

(2) If $\sigma_2 > \frac{\sigma_1 + \sigma_3}{2}$, according to Eq.(3):

$$\sigma_{UE} = \frac{1}{1+b} (\sigma_1 + b\sigma_2) - \sigma_3 = \sigma_u \quad (12)$$

By substituting Eqs. (1) and (5) into Eq. (16), it can be obtained that:

$$\frac{1}{1+b} ((\sigma_L)_p + \Sigma (\sigma_L) + b * p \frac{D_e}{2t}) = \sigma_u \quad (13)$$

$$\frac{1}{1+b} \left(\frac{(\sigma_L)_p}{\sigma_u} + \frac{\Sigma(\sigma_L)}{\sigma_u} + b * \frac{p \frac{D_e}{2t}}{\sigma_u} \right) = 1 \quad (14)$$

Substitute Eq. (2) and (6) into Eq. (18), it can be obtained that:

$$\frac{1}{1+b} \left(\frac{\frac{1}{2} * p \frac{D_e}{2t}}{p_0 \frac{D_e}{2t}} + \frac{\Sigma(\sigma_L)}{\sigma_u} + b * \frac{p \frac{D_e}{2t}}{p_0 \frac{D_e}{2t}} \right) = 1 \quad (15)$$

$$\frac{1}{1+b} \left(\frac{1}{2} * \frac{p}{p_0} + \frac{\Sigma(\sigma_L)}{\sigma_u} + b * \frac{p}{p_0} \right) = 1 \quad (16)$$

$$\left(\frac{1}{2(1+b)} + \frac{b}{b+1} \right) \frac{p}{p_0} + \frac{1}{1+b} \frac{\Sigma(\sigma_L)}{\sigma_u} = 1 \quad (17)$$

This is the derivation process of Eq. (17) in the old version manuscript (Eq. (17) in this document). In order to prevent readers from misunderstanding the theoretical derivation process, the manuscript has been supplemented with the followings:

“

By substituting Eqs. (13) and (6) into Eq. (19), it can be obtained that:

$$\frac{1}{1+b} \left((\sigma_L)_p + \Sigma(\sigma_L) + b * p \frac{D_e}{2t} \right) = \sigma_u \quad (20)$$

$$\frac{1}{1+b} \left(\frac{(\sigma_L)_p}{\sigma_u} + \frac{\Sigma(\sigma_L)}{\sigma_u} + b * \frac{p \frac{D_e}{2t}}{\sigma_u} \right) = 1 \quad (21)$$

Substitute Eqs. (6), (7) and (14) into Eq. (21), it can be obtained that:

$$\frac{1}{1+b} \left(\frac{\frac{1}{2} * p \frac{D_e}{2t}}{p_0 \frac{D_e}{2t}} + \frac{\Sigma(\sigma_L)}{\sigma_u} + b * \frac{p \frac{D_e}{2t}}{p_0 \frac{D_e}{2t}} \right) = 1 \quad (22)$$

The limit state equation is defined as:

$$\left(\frac{1}{2(1+b)} + \frac{b}{b+1} \right) \frac{p}{p_0} + \frac{1}{1+b} \frac{\Sigma(\sigma_L)}{\sigma_u} = 1 \quad (23)$$

”

The other limit state equations as Eqs. (24), (26), (27) and (28) are obtained with a

similar process, so they will not be repeated in the manuscript.

5. Comment: In section 4, what effect does the loading sequence have on the experimental results?.

Author's Response:

I fully agree with the reviewer's viewpoint that the loading path does indeed have an impact on the experimental results.

1) The relevant research results are as follows:

Reference 1: Smith, M.Q., Waldhart, C.J., 2000. Combined Loading Tests of Large Diameter Corroded Pipelines. The 3rd International Pipeline Conference, Calgary, Alberta, Canada. pp. V2T-V6T.

A multi-year combined testing and analysis program was initiated by the Alyeska Pipeline Service Company aimed at developing computer tools for the prediction of rupture and wrinkling in corroded pipes. During the program, seventeen full-scale tests of mechanically corroded 48-inch diameter (1219-mm), X65 pipes subjected to internal pressure, axial bending, and axial compression were performed to provide data necessary for the verification of analytical models and failure prediction models

The two initial Phase III tests were designed to evaluate load path sensitivity. During the first test (III-1), bending was applied in the initial load step. Pressure, with compensating axial loading was applied after the initial load step until rupture occurred in the compression side defect. In the second Phase III test (III-2), pressure with compensating axial load was applied until the rupture pressure of the III-1 specimen was achieved, followed by bending of the pipe until rupture was achieved, again within the compression side defect.

The results turn out that rupture occurred at virtually the same pressure and bending moment in each test. However, comparison of the the hoop strain and deflection variations show that while **the pressure and moment produced at rupture are almost equivalent, a very different mode of failure is obtained.**

Reference 2: Wang, W., Smith, M.Q., Popelar, C.H., Maple, J.A., 1998. A New Rupture Prediction Model for Corroded Pipelines Under Combined Loadings. The 2nd International Pipeline Conference, Calgary, Alberta, Canada. pp. 563-572.

Under combined effects of internal pressure, axial bending, thermal loading and other secondary loading, the rupture condition of a corroded pipe is dictated by the load path and the boundary condition. In a load controlled environment, secondary axial stresses may be significant at rupture. In a displacement controlled environment, however, the secondary axial stresses decrease as the pressure increases, and rupture takes place when the hoop strain reaches the limit value. **As long as there is sufficient strain capacity during the pressurization stage for the secondary axial stresses to vanish, the bending and other secondary loading on the pipe will not affect its rupture pressure.**

Reference 3: Paslay, P.R., Cernocky, E.P., Wink, R., 1998. Burst Pressure Prediction of Thin-Walled, Ductile Tubulars Subjected to Axial Load. SPE Applied Technology Workshop on Risk Based Design of Well Casing and Tubing, Woodlands, Texas. pp. 48327.

In order to develop some quantitative information concerning path dependence, BURST2 was modified and then applied to two load paths. The first is the path followed in the burst experiments where the external axial force F_{EXT} is applied initially and held constant while the pressure is increased to burst. The second load path is for the pressure P to be applied initially and held constant while the axial external force F_{EXT} is increased until plastic instability occurs. However, BURST2 was run in acceptable ranges for both paths with a 7-in. OD, 0.875-in. wall, K55 tubular. The result of these calculations is given in Fig. 6, where the solid curve corresponds to the first load path and the dashed line corresponds to the second load path. **The curves are within about 10% of one another and suggest that load path is not a major concern in burst design for ductile tubulars.** The Quantitative Risk Analysis (QRA) modeling did not take the influence of load path into account.

[REDACTED]

Fig. 6 Predictions of plastic instability for combined axial load and internal pressure based on two load paths.

Reference 4: Gao, Z., Cai, G., Liang, L., Lei, Y., 2008. Limit Load Solutions of Thick-Walled Cylinders with Fully Circumferential Cracks Under Combined Internal Pressure and Axial Tension. Nucl Eng Des. 238(9), 2155-2164.

To investigate the influence of loading sequence on the limit load of a cylinder under combined loads, an uncracked cylinder is analyzed using two loading sequences (Fig. 7), one is proportional loading, i.e. the internal pressure and the axial force are applied simultaneously with a fixed load ratio, along route 1 in Fig. 7, and another is sequent loading, i.e. the internal pressure is applied first (route 2 in Fig. 7) and then the axial force (route 3 in Fig. 7). The normalized limit loads obtained from the FE analyses for various values of load ratios are compared in Fig. 8.

From Fig. 7, it is seen that the normalized limit load values obtained from proportional loading and sequent loading are almost identical. **In other words, the loading sequence does not affect the limit load solution.**

[REDACTED]

Fig. 7. Loading paths.

[REDACTED]

Fig. 8. Comparison of the normalized FE limit load values between proportional and sequent loadings

2) The team of the manuscript authors' results

The team of the manuscript authors also evaluated the impact of loading sequence on ultimate bearing capacity, and the results are as follows :

The finite element method was used to evaluate the bearing capacity of corroded pipelines, and the calculated parameters are shown in Table 1. The FE model and true stress-strain curve are shown in Fig. 9 and 10.

Tab. 1 Case of analysis

grade	diameter D (mm)	Wall thickness t (mm)	corrosion depth d/t	corrosion length L^2/Dt	corrosion width θ / π
API-5L-X60	324	10	0.6	8.1	0.3

Fig. 9 Finite element model with corrosion defect

Fig. 10 True stress- strain relationship for API X60 pipeline steel

(1) Combined action of internal pressure and bending moment load

Apply internal pressure and bending moment loads in different orders to corroded pipelines with the same corrosion defect size and geometric size, and study their remaining ultimate bearing capacity. **Load sequence 1:** Apply internal pressure load to the corroded pipeline first, maintain a constant value after reaching a certain value, and then apply bending moment load to the pipeline until it is damaged. At this point, the corresponding bending moment load is the ultimate bending moment bearing capacity of the corroded pipeline. **Load sequence 2:** First, apply a bending moment load to the corroded pipeline, maintain it at a constant value, and then apply an internal pressure load to the pipeline until it is damaged. At this point, the corresponding internal pressure load is the ultimate internal

pressure bearing capacity of the corroded pipeline.

Fig. 11 shows a comparison of the ultimate bearing capacity of corroded pipelines under different loading sequences. According to Figure 3.2, it can be seen that the pipeline bearing capacity under the "bending moment internal pressure" loading sequence is lower than that under the "internal pressure bending moment" loading sequence, but **the difference between the two is small, with a maximum difference of 7.31%.**

Fig. 11 Ultimate bending moment capacity influenced by corrosion depth

(2) Combined action of internal pressure and axial force

Two different loading sequences were used to study the effect of loading sequence on the ultimate bearing capacity of corroded pipelines under the combined action of internal pressure and axial force. **Load sequence 1:** First, apply internal pressure load to the corroded pipeline, maintain it at a constant value, and then apply axial force load to the corroded pipeline until the pipeline is damaged, that is, the equivalent stress of the pipeline reaches the ultimate tensile strength. **Load sequence 2:** Apply axial force load to the corroded pipeline first, reach the specified value, maintain it as a constant, and then apply internal pressure load to the pipeline until it is damaged. At this point, the corresponding internal pressure load is the ultimate internal pressure bearing capacity of the corroded pipeline.

Fig. 12 shows a comparison of the ultimate bearing capacity of corroded pipelines

under different loading sequences. According to Fig. 12, it can be seen that the pipeline bearing capacity under the "internal pressure axial force" loading sequence is lower than that under the "axial force internal pressure" loading sequence, but **the difference between the two is small, with a maximum difference of 5.27%.**

Fig. 12 Ultimate capacity influenced by corrosion depth

3) Conclusion

It is well known both theoretically and from experiments that final states are dependent on load path for elastic–plastic materials loaded beyond their yield points. This path dependence could be a complication in the burst design of pipelines. The curves are within about 10% of one another and suggest that load path is not a major concern in burst design for ductile tubulars (Paslay et al., 1998). **Therefore, the experimental study in this manuscript ignores the influence of loading sequence on the ultimate internal pressure of pipelines.**

There are two main reasons for the experimental loading path in this manuscript:

(1) Since this work focuses on the burst pressure of the corroded pipelines, the loading path of the model is the external loads first and then internal pressure.

(2) The test method was based on a load path similar to that experienced by a pipeline in oil and gas, i.e. the axial force was applied to the pipe followed by internal pressure (Wang et

al., 1998, Smith et al., 1998).

(3) This load loading route has been widely adopted in literature (Wang et al., 1998, Smith et al., 1998, Lasebikan and Akisanya, 2014, Paslay et al., 1998, Liu et al., 2009, Grigory and Smith, 1996).

Reference

- Paslay, P.R., Cernocky, E.P., Wink, R., 1998. Burst Pressure Prediction of Thin-Walled, Ductile Tubulars Subjected to Axial Load. SPE Applied Technology Workshop on Risk Based Design of Well Casing and Tubing, Woodlands, Texas. pp. 48327.
- Wang, W., Smith, M.Q., Popelar, C.H., Maple, J.A., 1998. A New Rupture Prediction Model for Corroded Pipelines Under Combined Loadings. The 2nd International Pipeline Conference, Calgary, Alberta, Canada. pp. 563-572.
- Smith, M.Q., Nicolella, D.P., Waldhart, C.J., 1998. Full-Scale Wrinkling Tests and Analyses of Large Diameter Corroded Pipes. The 2nd International Pipeline Conference, Calgary, Alberta, Canada. pp. 543-551.
- Lasebikan, B.A., Akisanya, A.R., 2014. Burst Pressure of Super Duplex Stainless Steel Pipes Subject to Combined Axial Tension, Internal Pressure and Elevated Temperature. Int J. Pres Ves Pip. 119, 62-68.
- Liu, J., Chauhan, V., Ng, P., Wheat, S., Hughes, C., 2009. Remaining Strength of Corroded Pipe Under Secondary (Biaxial) Loading. GL Industrial Services UK Ltd, Washington, DC, USA: US Department of Transportation.
- Grigory, S.C., Smith, M.Q., 1996. Residual Strength of 48-Inch Diameter Corroded Pipe Determined by Full Scale Combined Loading Experiments. 1st International Pipeline Conference, Calgary, Alberta, Canada. pp. 377-386.

6. Comment: In section 5.2, it is best to introduce the finite element model in detail, such as modeling, mesh division, boundary conditions, etc.

Author's Response:

The details of the FE analysis model is provided, which is as follows:

“Establish a geometric model of the pipeline with ANSYS. Establish a finite element model of the pipeline using a three-dimensional solid unit C3D8R. Divide the pipeline into 4 layers of units in the thickness direction, 48 units in the circumferential direction, and 88 units in the axial direction. Set reference points at both ends of the pipeline, and use rigid beam constraints between the reference points and the pipeline end nodes. The analyzed model is as shown in Fig. 11.

Fig. 11. FE model for predicting the burst pressure

By applying the bending moment (M) and axial force (F) simultaneously to the reference node, the internal pressure on the inner surface nodes was increased gradually until the pipe experienced burst failure.

To solve the finite element model, the nonlinear arc-length method algorithm was employed. The simulation utilized a pipeline material exhibiting isotropic hardening plasticity. For the true stress-strain relationship, the well-established Ramberg-Osgood model was adopted.

The burst data of Table 4 was adopted to to verify the correctness of the model and the results are listed in Table 5. It illustrates that the average error is 3.11%, which is within the acceptable level. The accuracy of the model has been verified.

Table 4 The errors of the FE model

steel grade	case	failure pressure in the test (MPa)	failure pressure with FE model (MPa)	error
Q235	test in the paper	11.69	11.70	0.09%
L80	13	115.48	111.67	-3.30%
L80	16	64.08	68.68	7.18%
L80	17	50.64	50.66	0.03%
K55	18	54.84	57.37	4.62%
Q125	19	63.39	61.20	-3.45%
	min		-3.45%	
	max		7.18%	
	average		3.11%	
	Standard deviation		0.0427	

Note: error = $(P_{FE} - P_T) / P_T \times 100\%$ · P_{FE} is the predicted value, P_T is the failure pressure of test. average = $\sum |error| / 6$

The size and strength data of X52 in the Yi Shuai and Xiao Zhang et. al. models were adopted for analysis (Shuai et al., 2022). The reasons for adopting the data are as follows:

(3) Due to the lack of pipeline data with medium strength in the burst test, X52 steel was adopted as a representative for analysis.

(4) At present, there is a lack of data on the large diameter to thickness ratio in the test (the diameter to thickness ratio of burst test in Table 4 are 14.5~73.0), therefore $D/t=96$ is adopted as a representative.

The specific finite element model is shown in Tab. 5.”

7. Comment: In conclusions, “(2) Circumferential stress remains a key indicator of internal pressure failure in pipelines with complex loads. Tensile stress will increase the internal pressure bearing capacity of the pipeline, while compressive stress has the opposite effect” Conclusion 2 conflicts with the experimental data in the literature (Lasebikan B A, Akisanya

A R. Burst pressure of super duplex stainless steel pipes subject to combined axial tension, internal pressure and elevated temperature[J]. International Journal of Pressure Vessels and Piping, 2014, 119: 62-68.) According to Lasebikan's experiments, axial tension, regardless of the value, will reduce the failure pressure of the pipe. Please check.

Author's Response:

It should be noted that in the experimental results of Lasebikan and Akisanya (Lasebikan and Akisanya, 2014), the axial tension is the **effective axial stress**. That is the sum of the axial stresses, which consider both the axial stress generated by external axial loads and the axial stress generated by internal pressure.

The axial stress in the conclusion of this article refers to the **external axial stress**, which is only generated by the external axial force and does not consider the axial stress under internal pressure. Regarding the impact of external axial stress on the failure pressure of pipelines, relevant research is as follows:

1) Analysis results of existing literature

(1) A three-dimensional (3D) nonlinear finite element (FE) model was developed by Yi Shuai et al. (Shuai et al., 2022) to investigate the effect of bending moment and axial force on the burst capacity of corroded pipelines.

(Reference: Shuai, Y., Zhang, X., Feng, C., Han, J., Cheng, Y.F., 2022. A Novel Model for Prediction of Burst Capacity of Corroded Pipelines Subjected to Combined Loads of Bending Moment and Axial Compression. Int J. Pres Ves Pip. 196, 104621.)

□ The influence of external axial force generated by bending moment

In this article, Yi Shuai provides a comparison of failure pressures at different positions with bending moment action. The results are shown in Fig. 3 and Fig. 4. **It can be seen that the failure pressure on the tensile side (180°) is the highest, while the failure pressure on the compressive side (0° and 360°) is the lowest. The failure pressure on the neutral axis (90° and 270°) is between the two results**

[REDACTED]

Fig. 1. The circumferential position of defect expressed by the angle (θ) between centerline of the defect and the bending direction of the pipe.

[REDACTED]

Fig. 2. Effect of the location of corrosion defect on failure pressure of the pipe.

□ The influence of external axial force

The article also analyzed the magnitude and direction of the external axial stress, as shown in Fig. 3. **It can be seen that as the axial compressive stress increases, the failure pressure of the pipeline decreases; As the axial tensile stress increases, the failure pressure of the pipeline increases.**

The original content of the literature is as follows:

“Fig. 3 shows the relationship between the burst pressure and the bending moment under various axial forces. As a whole, the burst pressures of the pipes under different combined load conditions are smaller than 9.658 MPa, which is the burst pressure of the pipe subjected to internal pressure only. Moreover, the burst pressure decreases with the increased bending moment M/M_0 . The burst pressure is approximately linear related to the bending load. Furthermore, with a same bending moment value, the burst pressure of the pipe under an axial compression is smaller than the burst pressure under an axial tension. **The greater the axial compression force, the lower the burst capacity of the pipe. As the axial tensile load increases, the burst capacity of the pipe increases.**”

[REDACTED]

Fig. 3. Relationship between failure pressure of the corroded pipe and the bending moment under various axial forces.

(where F_0 is the buckling load and $F_0 = \pi D t \sigma_y$; M_0 is the ultimate elastic buckling moment: $M_0 = D^2 t \sigma_y$)

□ evaluation method

The article proposed a evaluation method for estimating the internal pressure bearing capacity, which is shown as Eq.(5).

$$p_f = p_0 \left(1 - 0.23578 \frac{F}{F_0} \right)^2 \quad (5)$$

(where p_f is the failure pressure, p_0 is the failure pressure of pipeline with only internal

pressure, the compression force is positive, negative tensile force.)

From Eq. (5), it can also be seen that for axial compression force, as the axial load increases, the failure pressure continuously decreases; For tensile stress, as the axial load increases, the failure pressure continuously increases.

(2) In order to study the influence of axial stress on ultimate internal pressure of corroded pipelines, a three-dimensional finite element model of pipeline was established by Zhao Pengcheng et al. (Pengcheng et al., 2019). The changes of ultimate internal pressure of pipelines under axial compressive stress and axial tensile stress with different corrosion defect parameters and material parameters were analyzed.

(Reference: Pengcheng, Z., Jian, S., Yu, T., Kui, X.U., 2019. Impact of Axial Stress On Ultimate Internal Pressure of Corroded Pipelines. *China Safety Science Journal*. 29(3), 70-75.)

The results show that the axial compressive stress will lead to reduction of the ultimate internal pressure, that the effect of axial stress on the ultimate internal pressure of pipelines varies and the smaller the defect size is, the more obvious the effect will be, **that the axial tensile stress will result in a slightly higher ultimate internal pressure, and the greater the axial stress is, the more obvious the effect will be.** The results are shown in Fig. 4 and Fig. 5 of the paper.

[REDACTED]

Fig.4 Influence of axial stress on ultimate internal pressure of pipes with different defect lengths

[REDACTED]

Fig.5 Influence of axial stress on ultimate internal pressure of pipes at different defect depths

2) Analysis results of burst test

Circumferential strain remains a key indicator of internal pressure failure. The variation of circumferential strain under internal pressure and axial tensile stress is shown in Fig. 6.

Axial tension can cause a reduction in the cross-section of the pipeline, resulting in circumferential compressive stress. And internal pressure causes pipeline damage through circumferential tension, so circumferential compressive stress can increase the internal pressure bearing capacity of the pipeline.

In order to overcome the circumferential compressive strain caused by axial tensile stress, an additional internal pressure Δp needs to be applied. The circumferential stress of the pipeline is only generated by internal pressure load. Assuming that the circumferential failure strain remains constant during pipeline burst with different load combinations, so $p_f = \Delta p + p_0$, which is greater than the failure pressure p_0 with only internal pressure.

(a) test data

(b) sketch figure

Fig. 6 Hoop strain curve

However, this conclusion contradicts the results of some published articles, most of which believe that both axial tension and axial compressive stress can cause a decrease in the internal pressure bearing capacity of pipelines. Such as: Phan, H.C., et al. (Phan et al., 2021), Bhardwaj, U. et al. (Bhardwaj et al., 2022), Lasebikan, B.A. and Akisanya, A.R. (Lasebikan and Akisanya, 2014), Mondal, B.C. and Dhar, A.S. (Mondal and Dhar, 2019) and Paslay, P.R. et al. (Paslay et al., 1998).

The conclusion of this article on the effect of axial tension on failure pressure is

qualitatively derived based on the variation law of strain data in the test and **the maximum principal strain failure criterion (which is shown as Fig. 6)**. After referring to relevant literature, it is believed that this conclusion has not been widely recognized and the conclusion is not the core content of the manuscript. Therefore, the conclusion is revised as follows:

“(2) Circumferential stress remains a key indicator of internal pressure failure in pipelines with complex loads. **The maximum circumferential failure strain is approximately 3.43 times the maximum axial failure strain.**”

Reference

- Phan, H.C., Le, T., Bui, N.D., Duong, H.T., Pham, T.D., 2021. An Empirical Model for Bending Capacity of Defected Pipe Combined with Axial Load. *Int J. Pres Ves Pip.* 191, 104368.
- Bhardwaj, U., Teixeira, A.P., Soares, C.G., 2022. Failure Assessment of Corroded Ultra-High Strength Pipelines Under Combined Axial Tensile Loads and Internal Pressure. *Ocean Eng.* 257, 111438.
- Lasebikan, B.A., Akisanya, A.R., 2014. Burst Pressure of Super Duplex Stainless Steel Pipes Subject to Combined Axial Tension, Internal Pressure and Elevated Temperature. *Int J. Pres Ves Pip.* 119, 62-68.
- Mondal, B.C., Dhar, A.S., 2019. Burst Pressure of Corroded Pipelines Considering Combined Axial Forces and Bending Moments. *Eng Struct.* 186, 43-51.
- Paslay, P.R., Cernocky, E.P., Wink, R., 1998. Burst Pressure Prediction of Thin-Walled, Ductile Tubulars Subjected to Axial Load. *SPE Applied Technology Workshop on Risk Based Design of Well Casing and Tubing*, Woodlands, Texas. pp. 48327.

REVIEWER COMMENTS

Reviewer #1 (Remarks to the Author):

Thank you for revising the manuscript. I still have comments on your paper. Please check the attached file for detail comments.

Reviewer #2 (Remarks to the Author):

I recommend this paper for publication.

Paper ID: NCOMMS-23-43457

Paper Title: Research on the Limit State Equation and Failure Pressure Prediction Model of Pipeline with Complex Loading

[Comments]

I've checked the revised manuscript and the author's rebuttal.

I have to say that the authors tend to be wordy, and sometimes it obscures the important points what be focused. For example, regarding the testing procedure described in 4.3, the main purpose of procedure (3) is pressurizing. Water injection is not the point in procedure (3). But the authors tried to explain everything of procedure (3), and it makes me confused what the main purpose of this procedure is. In the introduction section, the authors describe the previous research on corroded pipe; however, the authors' research work is done on intact pipe. I cannot understand well why the authors devote much space to the research on corroded pipe. Though these examples are relatively minor problems, there are a lot of similar descriptions in the manuscript. I think the manuscript could be better organized.

In addition, the following points should be clarified.

(1) Table 3:

382

Table 3 Data of Burst test

literature Sources	test No.	diameter (mm)	grade	D/t	axial stress (MP)
	13	88.9	L80	14.5	348.7924
P.R. Paslay et al.	16	88.9	L80	18.0	586.6628
(Paslay et al., 1998)	17	88.9	L80	18.6	638.1449
	18	177.8	K55	26.7	454.3197
	19	196.85	Q125	18.1	884.7380
	22°C-1				450
	22°C-2				450
	22°C-3				460
	22°C-4				550
	22°C-5				540
	22°C-6				650
	22°C-7				655
	22°C-8				665
Lasebikan et al.	22°C-9				750
(Lasebikan and Akisanya,	22°C-10	8	125	0.25	751
	22°C-11				740

Diameter: 8 mm, D/t: 0.25???

(2) Lines 415-416: the authors explained that the FEM software was ANSYS, and the element type was C3D8R; however, in my understanding, "C3D8R" is the element type in ABAQUS, not ANSYS.

(3) The result that the von Mises criterion shows good agreement with the actual behavior is not so new finding. It is better to show concisely what the advantage of this paper is.

Date: Feb. 23, 2024

To: Nature Communications

From: SUN Ming-ming, FANG Hong-yuan*, WANG Nian-nian, DU Xue-ming, ZHAO Hai-sheng, Zhai Ke-Jie

Re: Manuscript No.: NCOMMS-23-43457

Title: Research on the Limit State Equation and Failure Pressure Prediction Model of Pipeline with Complex Loading

Dear reviewer, Thank you for reviewing our manuscript and for the constructive comments, which greatly helped us to improve the manuscript. We have heavily revised our experiments. Mainly revise the structure and discourse style of the article.. The manuscript was carefully revised and point-by-point response was listed below. We hope that your comments have been addressed accurately. The revised content in the manuscript has been **highlighted in yellow**.

Reviewer #1

1. Comment: I've checked the revised manuscript and the author's rebuttal.

I have to say that the authors tend to be wordy, and sometimes it obscures the important points what be focused. For example, regarding the testing procedure described in 4.3, the main purpose of procedure (3) is pressurizing. Water injection is not the point in procedure (3). But the authors tried to explain everything of procedure (3), and it makes me confused what the main purpose of this procedure is. In the introduction section, the authors describe the previous research on corroded pipe; however, the authors' research work is done on intact pipe. I cannot understand well why the authors devote much space to the research on corroded pipe. Though these examples are relatively minor problems, there are a lot of similar descriptions in the manuscript. I think the manuscript could be better organized.

Author's Response:

Thank you very much for the careful review of the manuscript by the reviewer. The author has made revisions to the writing structure and approach of the entire manuscript. The main modifications are as follows:

1) The testing procedure described in 4.3

The main purpose of procedure (3) is to apply pressure and record the pressure value, and water injection is only a means of applying pressure. The modified content is as follows:

“(3) The pressurization device applies pressure to the pipeline through the movement of the piston in the water tank. Measure the internal pressure load with high-precision pressure sensors, and transmit the sensor signal to the measurement control system to achieve feedback and record the internal pressure load value.”

2) Simplify the experimental process

Add Appendix A to discuss the experimental process, with a focus on the experimental results in the manuscript. At the same time, refine the sentences in the experimental introduction, such as:

DELETE “The characteristics of intact pipeline burst test are as follows: (1) Unable to predict the failure location and provide effective damage protection; (2) The constraints at both ends of the pipeline are prone to stress concentration and damage, which cannot accurately reflect the failure mode of the intact pipeline.”

“The closure and end fixation of the test is shown in Fig. A2. The end of the pipeline is closed with a flange plate, and there is a 20mm groove inside the flange plate to allow the pipeline to be inserted and provide better sealing effect. Circumferential welding shall be carried out at the connection between the pipeline and the flange plate. To enhance the connection between the pipeline and the flange plate, a stiffener is welded to the pipeline. There are water injection holes and outlet holes on the side of the flange, which are connected to the water supply pipe through joints and fixed and sealed with nuts. The flange plate has bolt holes, which are linked to the loading device through bolts for axial force loading.”

“The internal pressure loading device includes water injection, drainage, and pressurization devices. The pressurization device is a set of stainless steel boosting cylinders

driven by servo oil cylinders, with a stroke of 600 mm and the ability to apply a maximum pressure of 50 MPa. The control platform can control the pressure application method through sensors, including displacement control and load (pressure) control.”

3) The content of introduction

The analysis of the bearing capacity of intact pipelines is the basis for evaluating corroded pipelines. Currently, most researchers use finite element models or semi empirical formulas to evaluate the bearing capacity of pipelines. The product of the calculation model for the bearing capacity of intact pipelines and the reduction coefficient caused by corrosion defects is adopted as the bearing capacity model for corroded pipelines. Therefore, the research on the evaluation of the bearing capacity of corroded pipelines under complex loads includes the evaluation of intact pipelines.

Considering that this manuscript mainly focuses on the assessment of the bearing capacity of intact pipelines, the introduction section has been revised to focus on the progress of the assessment of intact pipelines. The revised content are as follows:

“Without considering the safety factor, the failure pressure p_f of intact pipeline is generated as follow:

$$p_f = \frac{2t\sigma_b}{D-t} H_1 \quad (1)$$

$$H_1 = 2\left(1 + \frac{\sigma_L}{\sigma_b}\right) \quad (2)$$

where D is the diameter; t is the wall thickness; σ_L is axial stress; σ_b is tensile strength.

A prediction model of determining the burst pressure of pipeline with axial load was proposed by Zhao et al.³⁰.

Zhou et al.³¹ found that the axial stress lowered the failure pressure of pipeline and a prediction formula for failure pressure based on vertical strain was established. However, it is applied only to the pipeline with small defect or intact pipeline. Chen et al.³² first proposed semi-empirical failure pressure equation of pipelines subjected to internal pressure and axial tension, which is an improvement on the theory of evaluating the bearing capacity of pipelines with complex loads.

Chauhan and Swankee³³ and Liu et al.³⁴ proposed interactive charts to estimate the residual strength of pipeline with external load.

A mathematical programming formulation is presented by Heitzer³⁵ and the numerical procedure is applied to analyze the plastic collapse of pipelines with internal pressure and axial force.

Shuai et al.⁴⁰ investigated the buckling bearing capacity of pipeline with axial compressive loading and internal pressure by finite element analysis. Plastic collapse assessment procedure was proposed by Konosu and Mukaimachi⁴¹ for pressurized vessel with bending moment. Mohd et al.⁴² also investigated the same case of pipeline.

Gao et al.⁴⁴ analyzed the bending bearing capacity of pipelines with multiple loads.

Smith and Grigory⁵⁰ proposed the global buckling failure envelopes of pipeline accommodating combined stresses.

A parametric study of pipeline with bending moment and pull force was conducted by Agarwal⁵² and the optimization method of thickness distribution around the pipeline was proposed.”

4) The structure of introduction

Adjust the structure of the preface, and the content of different paragraphs in introduction after adjustment is as follows:

The first paragraph:

The importance of evaluating the internal pressure of steel pipelines is discussed.

The second paragraph:

Based on literature, it is explained that bending moment or axial force can reduce the internal pressure bearing capacity of pipelines, thereby highlighting the importance of internal pressure evaluation of pipelines with complex loads.

The third paragraph:

Analyze the research report on the internal pressure bearing capacity of pipelines with axial force and internal pressure.

The fourth paragraph:

Analyze the relevant research literature on the internal pressure bearing capacity of

pipelines with axial force and internal pressure.

The fifth Paragraph:

Other types of evaluation schemes for the bearing capacity of pipelines with axial force and internal pressure, in addition to mathematical calculation models

The sixth paragraph:

Discuss the literature on the evaluation of the bearing capacity with bending moment load or the combined action of bending moment and axial force, which mostly adopts experimental discussion and numerical simulation methods.

The seventh Paragraph:

Summarize the research content of relevant literature, point out the shortcomings of current research, and introduce the research content of this article.

5) Revision and refinement of English writing in manuscripts

The manuscript has been reviewed by English native language researchers in the relevant field. The main modifications are as follows:

“As a safe and economical material transportation carrier, steel pipelines are widely used in long-distance water diversion, urban heating, and the transportation of oil, gas, and other materials due to their large capacity, high bearing capacity and small impact on the environment¹⁻³. The size of pipeline and efficiency of material transportation are determined by the internal pressure⁴⁻⁶.”

“Buried pipelines are subjected to the combined action of various loads during operation⁷⁻⁹, including axial force, internal pressure, bending moment, etc. Internal pressure in buried pipelines is primarily due to the internal transport medium, with axial force originating from temperature differences during laying and operation.”

“ Geotechnical disturbances can cause axial forces to occur inside the pipeline. For example, when the pipelines cross a slope, the pipelines at the bottom of the slope will withstand axial compression and pipelines at the top will withstand axial tension¹⁰.”

“Taylor et al.¹² proved that the additional bending load decreased the failure pressure. For pipelines withstanding axial force, similar conclusions could also be drawn^{13,14}. Therefore, the assessment of failure pressure taking into account internal pressure only is limited, and it

is necessary to analyze the burst pressure of pipelines with complex loads.”

“The early research mainly focused on the bearing capacity evaluation of pipelines with two types of loads: axial force and internal pressure. Some literatures¹⁵⁻²² refer to the research project of the Southwest Research Institute (SwRI) conducted in the 1990s.”

“In order to compensate for the shortcomings of existing evaluation models, more complex failure pressure assessment schemes have been proposed.”

“A new failure pressure evaluation method was proposed by Benjamin³⁶ and Bruère et al.³⁷, which can evaluate the impact of axial compressive force.”

“The marine pipelines are modeled using the finite element method and its integrity is evaluated by Mondal and Dhar²⁹.”

DELETE“However, the axial deformation of buried pipelines is constrained by soil friction or the characteristics of the structural system, so the Poisson effect will cause axial tension inside the pipeline.”

“Other references²³⁻²⁸ refer to the "Reliability of corroded pipes" in the Joint Industry Projects (JIP) project developed by DET NORSKE VERITAS (DNV).”

“Zhou et al.³¹ found that the axial stress lowered the failure pressure of pipeline and a prediction formula for failure pressure based on vertical strain was established.”

“So far, the limit state model for pressurized pipelines with bending moment and axial force is still absent. Herein, the limit state equation of pipelines with complex loads is proposed based on the UYC (unified yield criterion) theory.”

“where σ_1 , σ_2 and σ_3 are the principal stresses and $\sigma_1 \geq \sigma_2 \geq \sigma_3$.”

The UYC is a set of different criteria as shown in Fig. 1⁵⁶. The value range of b is: $0 \leq b \leq 1$.”

Fig. 1 Yield criteria on the π -plane”

“Substituting Eqs. (6), (7) and (14) into Eq. (16) obtains:”

“Substituting Eqs. (6), (7) and (14) into Eq. (21) can obtain:”

“ The strain growth rate of R1-R12 is roughly the same and the pipeline exhibits overall uniform expansion. During the later loading stage as the internal pressure is greater than 3.4 MPa, the strain change rate at each measuring point begins to differ due to the concentrated deformation at the failure location.”

“Herein, a geometric model of the pipeline is established by ABAQUS and the corresponding finite element model is constructed using a three-dimensional solid unit C3D8R.”

2. Comment: Table 3:

382

Table 3 Data of Burst test

literature Sources	test No.	diameter (mm)	grade	D/t	axial stress (MF)
	13	88.9	L80	14.5	348.7924
P.R. Paslay et al. (Paslay et al., 1998)	16	88.9	L80	18.0	586.6628
	17	88.9	L80	18.6	638.1449
	18	177.8	K55	26.7	454.3197
	19	196.85	Q125	18.1	884.7380
	22°C-1				450
	22°C-2				450
	22°C-3				460
	22°C-4				550
	22°C-5				540
	22°C-6				650
	22°C-7				655
	22°C-8				665
Lasebikan et al. (Lasebikan and Akisanya,	22°C-9				750
	22°C-10				751
	22°C-11				740

Author’s Response:

I am very sorry for the inconvenience caused by the error in data organization. The wall thickness $t=0.25$, not D/t and the $D/t=32$. I have made corrections to Table 3 and verified other relevant data.

3. Comment: Lines 415-416: the authors explained that the FEM software was ANSYS, and the element type was C3D8R; however, in my understanding, “C3D8R” is the element type in ABAQUS, not ANSYS.

Author’s Response:

I am very sorry that the author's writing mistake has caused you inconvenience. Due to the superiority of the finite element software ABAQUS in nonlinear calculations, the manuscript uses ABAQUS for calculations. The text has been revised.

4. Comment: The result that the von Mises criterion shows good agreement with the actual behavior is not so new finding. It is better to show concisely what the advantage of this paper is.

Author's Response:

At present, there are many evaluation models for the bearing capacity of pipelines with individual internal pressure loads, but there are few models for evaluating the failure pressure of pipelines under complex loads, and the accuracy needs to be verified. The failure pressure assessment models for pipelines with multiple loads are mainly divided into the following four types:

1) Evaluation methods provided by specifications or reports

□ The widely used standard DNV-RP-F101 (DNV-RP-F101, 2015) provides a burst model for corroded pipelines, considering the effect of external loads by modifying the failure pressure model through an external compressive longitudinal stress factor H_1 .

The burst pressure p_f , is obtained by:

$$p_f = \frac{2t\sigma_b}{D-t} \left(\frac{1-(d/t)}{1-(d/t)/Q} \right) H_1 \quad (1)$$

$$H_1 = \frac{1 + \frac{\sigma_L}{\sigma_b A_r}}{1 - \frac{1}{2A_r} \frac{1-(d/t)}{1-\frac{(d/t)}{Q}}} \quad (2)$$

where D is the diameter; d and w are the depth and width of corrosion defect; t is the wall thickness; Q is the correction factor for corrosion length; A_r is the circumferential area reduction factor: $A_r = 1 - \frac{dw}{\pi Dt}$; σ_L is axial stress; σ_b is tensile strength. This model is suitable especially as the corrosion defect is withstand compressive stress. **However, as additional longitudinal stresses in the corrosion region $\sigma_L < 0.25\sigma_b$, the prediction accuracy remarkably decreased.**

□ Chauhan (Chauhan and Swankie, 2015) and Liu et al. (Liu et al., 2009) proposed interaction diagrams to evaluate the ultimate internal pressure of corroded pipelines under a single bending load or axial force. **However, this method cannot be applied on pipelines**

that are subject to a simultaneous loading of both bending moment and axial force.

2) The evaluation methods provided by using finite element data fitting

□ A prediction model was proposed for determination of burst capacity of corroded pipelines under an axial stress by Pengcheng et al. (Pengcheng et al., 2019). **However, the model was not applicable to a non-uniform axial stress caused by bending moment since it was developed based on the assumption that the cross-section of the pipe was under the same axial stress.**

□ Zhou et al. (Zhou et al., 2018) investigated the effect of a longitudinal strain caused by bending load on the burst capacity of a corroded pipeline by full-scale burst testing, and found that the longitudinal strain at defective area reduced the burst pressure. Based on FE analysis, a strain-based burst pressure prediction formula for pipelines containing a corrosion defect was developed. **However, the model was applicable for small defects only.**

□ A three-dimensional (3D) nonlinear finite element (FE) model was developed by Yi Shuai et al. (Shuai et al., 2022) to investigate the effect of bending moment and axial force on the burst capacity of corroded pipelines. Then based on a series of FE cases, a new burst prediction model for corroded pipelines subjected to the combination loads of bending load and axial compressive force was fitted and developed.

However, the finite element model and prediction formula have not been verified by intact pipeline failure experiments with both axial force and bending moment. And the evaluation model was fitted based on numerical results and did not theoretically derive an evaluation model for the internal pressure bearing capacity of pipelines under complex loads

□ Considering the axial compression and closing bending moment, failure loci of combined bending moments and internal pressures are developed for different axial forces by Mondal and Dhar (Mondal and Dhar, 2019). The developed failure loci can be used for assessing the burst pressure of corroded pipelines subjected to axial forces and bending moments. **For cases where the failure loci is not presented, the applicability of this method is reduced.**

3) Evaluation method with computational procedures

① Conventional work-hardening plasticity theory was used to establish a computer program for the prediction of the burst pressure in the presence of an axial load by Paslay et al. (Paslay et al., 1998). **However, the program was run extensively with the stress–strain curve, and does not provide a simple failure pressure prediction formula. This program is not applicable to the evaluation of pipeline failure pressure under bending moment action.**

□ A detailed analysis of the rupture and necking of tubulars is provided by Klever (Klever, 2006) and Stewart et al. (Stewart et al., 1994). The minimum internal pressure necessary for rupture is given, assuming von Mises yield criterion. **However, the axial effective stress was evaluated using this method and the influence of external axial stress magnitude and direction on the failure pressure was not directly given. And the axial effective stress changes with the magnitude of internal pressure, so it cannot be directly calculated for failure pressure.**

$$p_f = \frac{2}{(\sqrt{3})^{n+1}} \frac{2t\sigma_b}{D-t} \sqrt{1 - \frac{4^{1-n} - 1}{3^{1-n}} \left(\frac{\sigma_{eff}}{\sigma_b} \right)} \quad (3)$$

Where σ_{eff} is the effective stress in axial direction.

4) Evaluation methods of semi-empirical equation

Chen et al. (Chen et al., 2021) first proposed semi-empirical failure pressure equation of corroded pipelines subjected to internal pressure and axial tension, which is an improvement on the theory of evaluating the bearing capacity of pipelines with complex loads. **However, this method does not consider bending moment loads, and its accuracy remains to be confirmed for pipelines with other types of defects.**

Compared to existing evaluation methods, the evaluation method proposed in this article has the **following advantages:**

1) The existing evaluation methods are mainly divided into two types: one is based on the improvement of the bearing capacity evaluation model with the single load (internal pressure), and the other is based on the fitting of numerical calculation results. **The existing**

evaluation methods do not provide a limit state equation for pipelines from the perspective of mechanical mechanisms, which is not conducive to theoretically analyzing the impact of load size and type on the internal pressure bearing capacity of pipelines.

2) The existing evaluation models do not consider the impact of differences in yield criteria on the evaluation results. **The evaluation model proposed in the article is an integration of multiple yield criteria, and can be transformed into an evaluation model based on a single yield criterion by changing parameter b .** Therefore, this model can compare and analyze the evaluation results of different yield criteria, and thus clarify the optimal yield criteria applicable to different load modes.

3) Different application methods of external loads can affect the evaluation results of failure pressure, such as compressive force reducing the internal pressure bearing capacity of pipelines, while axial tension increasing the internal pressure bearing capacity. **The method proposed in the article proposes a corresponding evaluation model based on the differences in the distribution of principal stresses (which is shown in Table 1 in the manuscript).** Compared to the unity of existing evaluation methods, it has a wider applicability.

4) **The analysis method and theory proposed in the article have broader application prospects, and can be converted into bearing capacity evaluation of different failure modes based on changes in equivalent stress σ_{UE} (as shown in Eq. (4)).** For example, for internal pressure or axial tensile failure, the ultimate tensile strength can be selected as equivalent stress σ_{UE} for evaluation; For buckling failure, equivalent stress σ_{UE} can be adopted as evaluate buckling stress; For plastic failure, the equivalent stress σ_{UE} can be adopted as the yield stress for evaluation.

$$\sigma_{UE} = \begin{cases} \sigma_1 - \frac{1}{1+b}(b\sigma_2 + \sigma_3) & , \quad \sigma_2 \leq \frac{\sigma_1 + \sigma_3}{2} \\ \frac{1}{1+b}(\sigma_1 + b\sigma_2) - \sigma_3 & , \quad \sigma_2 > \frac{\sigma_1 + \sigma_3}{2} \end{cases} \quad (4)$$

Therefore, the main innovation of the manuscript lies in: firstly, the limit state

equation considers the effects of different loads and coupling modes; secondly, the state equation can be transformed into the failure pressure evaluation model with different yield criteria. The abstract in the manuscript has been revised as follows:

“Abstract: The limit state equation of steel pipes with different loads is the basis for evaluating the ultimate bearing capacity of pipelines. For buried pipelines, the most common operating loads are bending moment, internal pressure and axial force. Based on the three-dimensional stress model of pipelines with complex load coupling, the limit state equation of intact pipelines is proposed. The limit state equation can be transformed into a series of failure pressure evaluation models for pipeline with different yield criteria. Based on the failure data of pipelines, the applicability of different yield criteria in evaluating the failure pressure of pipelines with complex loads was analyzed.”

Reference

- DNV-RP-F101, 2015. Recommended Practice DNV-RP-F101 Corroded Pipelines. Det-Norske-Veritas, Norway.
- Chauhan, V., Swankie, T., 2015. Guidance for Assessing the Remaining Strength of Corroded Pipelines. Pipeline and Hazardous Materials Safety Administration, Washington, DC, USA: US Department of Transportation.
- Liu, J., Chauhan, V., Ng, P., Wheat, S., Hughes, C., 2009. Remaining Strength of Corroded Pipe Under Secondary (Biaxial) Loading. GL Industrial Services UK Ltd.
- Pengcheng, Z., Jian, S., Yu, T., Kui, X.U., 2019. Impact of Axial Stress On Ultimate Internal Pressure of Corroded Pipelines. China Safety Science Journal. 29(3), 70-75.
- Zhou, H., Wang, Y., Stephens, M., Bergman, J., Nanney, S., 2018. Burst Pressure of Pipelines with Corrosion Anomalies Under High Longitudinal Strains. 12th International Pipeline Conference, Calgary, Alberta, Canada. pp. V2T-V6T.
- Shuai, Y., Zhang, X., Feng, C., Han, J., Cheng, Y.F., 2022. A Novel Model for Prediction of Burst Capacity of Corroded Pipelines Subjected to Combined Loads of Bending Moment and Axial Compression. Int J. Pres Ves Pip. 196, 104621.
- Mondal, B.C., Dhar, A.S., 2019. Burst Pressure of Corroded Pipelines Considering Combined Axial Forces

and Bending Moments. Eng Struct. 186, 43-51.

Paslay, P.R., Cernocky, E.P., Wink, R., 1998. Burst Pressure Prediction of Thin-Walled, Ductile Tubulars Subjected to Axial Load. SPE Applied Technology Workshop on Risk Based Design of Well Casing and Tubing, Woodlands, Texas. pp. 48327.

Klever, F.J., 2006. Formulas for Rupture, Necking, and Wrinkling of Octg Under Combined Loads. SPE Annual Technical Conference and Exhibition, San Antonio, Texas, U.S.A. pp. 102585.

Stewart, G., Klever, F.J., Ritchie, D., 1994. An Analytical Model to Predict the Burst Capacity of Pipelines. 13th International Offshore Mechanics and Arctic Engineering Conference, Houston, Texas. pp. 177-188.

Chen, Z., Wang, W., Yang, H., Yan, S., Jin, Z., 2021. On the Effect of Long Corrosion Defect and Axial Tension On the Burst Pressure of Subsea Pipelines. Appl Ocean Res. 111, 102637.

REVIEWERS' COMMENTS

Reviewer #1 (Remarks to the Author):

Thank you for submitting the revised manuscript. I've confirmed that the manuscript has been updated adequately.

I have some minor comments to the manuscript. Please revise the manuscript as necessary.

(1) Line 54:

1) "Projects (JIP) project developed by ..." -> "Projects (JIP) developed by ..."

2) "The method modifies ..." Please indicate what "the method" here means.

(2) Line 141: "... and axial stress (σ_L)_p can be ..." -> "... and axial stress (σ_L)_p by pressure can be ..."

(3) 4.2.1 Calculation of axial force: it is better to mention the calculated value (which is I think adopted in the experiment).

(4) Figs 7(a) and 7(b): it is better to use the same line properties for the strain gages those were attached at the same position. (For example, it is better to plot "R14" by light-green solid line, as same as "R2".)

(5) Table 5: please explain how the authors determined the failure pressure in FE analysis in this table.

(6) Line 442, "-6000 kN – 2000 kN.": Does it mean that the loading capacity of the testing machine is 6000 kN in compression and 2000 kN in tension?

(7) Line 459, "a loading rate of 0.25 MPa/min": Does it mean that the pressure increasing rate was 0.25 MPa/min?

Date: Mar. 23, 2024

To: Nature Communications

From: SUN Ming-ming, FANG Hong-yuan*, WANG Nian-nian, DU Xue-ming, ZHAO Hai-sheng, Zhai Ke-Jie

Re: Manuscript No.: NCOMMS-23-43457C

Title: Research on the Limit State Equation and Failure Pressure Prediction Model of Pipeline with Complex Loading

Dear reviewer, Thank you for reviewing our manuscript and for the constructive comments, which greatly helped us to improve the manuscript. We have heavily revised our experiments. Mainly revise the structure and discourse style of the article.. The manuscript was carefully revised and point-by-point response was listed below. We hope that your comments have been addressed accurately. The revised content in the manuscript has been **highlighted in yellow**.

Reviewer #1

- 1. Comment:** 1) “Projects (JIP) project developed by ...” -> “Projects (JIP) developed by ...”.
2) “The method modifies ...” Please indicate what “the method” here means.

Author’s Response:

- 1) “Projects (JIP) project developed by ...” -> “Projects (JIP) developed by ...”.

Thank you very much for the careful review of the manuscript by the reviewer. The author checked the writing content of the entire manuscript. The modifications are as follows:

“Other references¹⁻⁶ refer to the "Reliability of corroded pipes" in the Joint Industry Projects (JIP) developed by DET NORSKE VERITAS (DNV).

- 2) “The method modifies ...” Please indicate what “the method” here means.

Thank you very much for the careful proofreading by the reviewer. The method was proposed by DNV as discussed in the previous sentence. In order to prevent readers from

misunderstanding, additional explanations have been provided in the article. The modifications are as follows:

“The method provided by DNV modifies the failure pressure model with the stress factor H_1 .”

2. Comment: Line 141: “... and axial stress $(\sigma_L)_p$ can be ...” -> “ ... and axial stress $(\sigma_L)_p$ by pressure can be ...”

Author’s Response:

Revised according to the reviewer's suggestions and the specific modifications are as follows:

“For thin-walled pipelines ($(D_e/t) \geq 20$), hoop stress σ_h and axial stress $(\sigma_L)_p$ by pressure can be obtained as Eqs. (6) and (7).”

3. Comment: 4.2.1 Calculation of axial force: it is better to mention the calculated value (which is I think adopted in the experiment).

Author’s Response:

The calculated value of N_c is 68kN and which is adopted in the experiment. The relevant calculation parameters are shown in Table 1.

$$N_c = -AE\lambda\Delta T \quad (1)$$

Table 1 The relevant calculation parameter

diameter	average wall thickness	E	λ	ΔT
154.6mm	2.11mm	2.11GPa	$11.8 \times 10^{-6}/^\circ\text{C}$	27°C

The calculation result is explained in the manuscript, and the specific modifications are as follows:

“The calculation result of N_c is 68 kN, and this value was adopted in the experiment.”

4. Comment: (4) Figs 7(a) and 7(b): it is better to use the same line properties for the strain gages those were attached at the same position. (For example, it is better to plot “R14” by

light-green solid line, as same as “R2”.)

Author’s Response:

Thanks very much for the reviewer's suggestions. Your suggestions have greatly helped improve the quality of the article's illustrations. As requested by the reviewer, curves of the same color and style have been used to draw the circumferential and axial strains at the same position. The modified content is as follows:

Fig. 7 Strain curve of intact pipeline

5. Comment: Table 5: please explain how the authors determined the failure pressure in FE analysis in this table.

Author’s Response:

By applying the bending moment (M) and axial force (F) simultaneously to the reference node, the internal pressure on the inner surface nodes was increased gradually until the pipe experienced burst failure. As the FE model established in this study is based on the mechanics of a continuous medium, it is not possible to simulate the development of discontinuities in the material. Thus, it is necessary to define a criterion for defining the failure pressure of the corroded pipe. The "residual wall thickness stress criterion" is adopted as the failure criterion⁷. That is as the maximum von Mises stress of the inner wall of the pipeline exceeds the true ultimate tensile strength, the pipe is damaged^{8,9}. The criterion is based on three assumptions: (1) As the residual wall thickness starts to necking, the corroded pipeline will experience local

plastic instability failure; (2) The stress state before necking is related to von Mises stress in the defect area¹⁰.

The "residual wall thickness stress standard" is originally developed according to the burst test database of corroded pipeline below grade X65⁷. Andrade et al.^{8,9} applied the same criteria to the nonlinear finite element analysis of API 5L X80 pipeline. The results show that the "residual wall thickness stress criterion" can be used in pipelines made of X60~X80 steels.

An excessive axial compression load and bending moment applied on the pipeline could cause local buckling. However, this work in the manuscript focused on burst failure dominated by circumferential stress. Thus, the axial force F and bending moment M applied in this study were controlled at low levels. It is assumed that the axial compressive force F and bending moment M are less than $0.5\sigma_u$ and $0.5M_0$, respectively, where M_0 is the critical bending moment of buckling for an intact pipe. This ensures that the longitudinal stress is smaller than the circumferential stress. Under actual conditions such as landslide, land subsidence and pipe in suspension, the external load applied on the pipeline is usually small. It is noted that, each FE modelling case has been evaluated by the widely used method of a load displacement curve¹¹⁻¹³, and no buckling occurred before a burst (That is, before the maximum Von Mises stress in the local defective area arrives the tensile strength, the load-displacement curve of the pipeline was not unloaded). Therefore, the local buckling due to longitudinal compression was not considered.

According to the reviewer's comments, add the method for determining the failure pressure of the finite element model in Table 5. The modified content is as follows:

Table 5 The errors of the FE model

steel grade	case	failure pressure in the test (MPa)	failure pressure with FE model (MPa)	error
Q235	test in the paper	11.69	11.70	0.09%
L80	13	115.48	111.67	-3.30%
L80	16	64.08	68.68	7.18%
L80	17	50.64	50.66	0.03%
K55	18	54.84	57.37	4.62%
Q125	19	63.39	61.20	-3.45%
	min		-3.45%	
	max		7.18%	
	average		3.11%	
	Standard deviation		0.0427	

Note: error = $(P_{FE} - P_T) / P_T \times 100\%$. P_{FE} is the predicted value, P_T is the failure pressure of test. Average = $\sum |\text{error}| / 6$. The "residual wall thickness stress criterion" is adopted for determining the failure pressure of pipelines^{6,60}.

References

1. Sigurdsson, G., Cramer, E. H., Bjørnøy, O. H., Fu, B. & Ritchie, D. Background to DNV RP-F101 Corroded pipelines. The 18th international conference on offshore mechanics and arctic engineering. 1999; Newfoundland, Canada: American Society of Mechanical Engineers; 1999.
2. Bjørnøy, O. H., Sigurdsson, G., Cramer, E. H., Fu, B. & Ritchie, D. Introduction to DNV RP-F101 Corroded Pipelines. The 19th International Conference on Offshore Mechanics and Arctic Engineering. 1999; Newfoundland, Canada: American Society of Mechanical Engineers; 1999.
3. Bjørnøy, O. H., Sigurdsson, G. & Marley, M. J. Background and development of DNV-RP-F101 corroded pipelines. *The Eleventh International Offshore and Polar Engineering Conference*. Stavanger, Norway: OnePetro; 2001. pp. 1-139.
4. Bjørnøy, O. H., Cramer, E. H. & Sigurdsson, G. Probabilistic Calibrated Design Equation For Burst Strength Assessment of Corroded Pipes. *The Seventh International Offshore and Polar Engineering Conference*. Honolulu, Hawaii, USA: OnePetro; 1997. pp. 160-166.
5. Bjørnøy, O. H., Sigurdsson, G. & Cramer, E. Residual strength of corroded pipelines, DNV test results. *The Tenth international offshore and polar engineering conference*. Seattle, Washington, USA: OnePetro; 2000. p. 140.
6. Bjørnøy, O. H. & Marley, M. J. Assessment of corroded pipelines: past, present and future. 11th International Offshore and Polar Engineering Conference. 2001 2001-01-01; Stavanger, Norway: International Society of Offshore and Polar Engineers; 2001. p. 1-138.
7. Batte, D., Fu, B., Kirkwood, M. G. & Vu, D. Advanced methods for integrity assessment of corroded

- pipelines. *Pipes and pipelines international*. **42**, 5-11 (1997).
8. Benjamin, A. C., de Andrade, E. Q., Jacob, B. P., Pereira, L. C. & Machado, P. R. Failure behavior of colonies of corrosion defects composed of symmetrically arranged defects. *International Pipeline Conference*. Calgary, Alberta, Canada: American Society of Mechanical Engineers; 2006. pp. 417-432.
 9. De Andrade, E. Q. et al. Finite element modeling of the failure behavior of pipelines containing interacting corrosion defects. 25th International Conference on Offshore Mechanics and Arctic Engineering. 2006; Hamburg, Germany: American Society of Mechanical Engineers Digital Collection; 2006. p. 315-325.
 10. Benjamin, A. C. & de Andrade, E. Q. Predicting the failure pressure of pipelines containing nonuniform depth corrosion defects using the finite element method. International Conference on Offshore Mechanics and Arctic Engineering. 2003 2003-01-01; Cancun, Mexico: American Society of Mechanical Engineers; 2003. p. 557-564.
 11. Shuai, Y., Wang, X. & Cheng, Y. F. Modeling of local buckling of corroded X80 gas pipeline under axial compression loading. *J. Nat. Gas Sci. Eng.* **81**, 103472 (2020).
 12. Shuai, Y., Wang, X. & Cheng, Y. F. Buckling resistance of an X80 steel pipeline at corrosion defect under bending moment. *J. Nat. Gas Sci. Eng.* **93**, 104016 (2021).
 13. Shuai, Y. et al. Local buckling failure analysis of high strength pipelines containing a plain dent under bending moment. *J. Nat. Gas Sci. Eng.* **77**, 103266 (2020).

6. Comment: Line 442, “-6000 kN – 2000 kN.”: Does it mean that the loading capacity of the testing machine is 6000 kN in compression and 2000 kN in tension?

Author’s Response:

Yes, “-6000 kN – 2000 kN” means that the loading capacity of the testing machine is 6000 kN in compression and 2000 kN in tension. The manuscript provides a clear explanation of the properties of force, such as “As the additional loadings is positive (tensile)”, “As the additional loadings is negative (compressive)” and “where N_c is a negative value and represents the axial compression force (N).”.

To prevent misunderstandings, the symbols for axial force are explained in the manuscript, and the revised content is as follows:

“The 6000 kN hydraulics cylinder is installed inside the pipeline base and can apply an axial force of -6000 kN (in compression)~2000 kN (in tension).”

7. Comment: Line 459, “a loading rate of 0.25 MPa/min”: Does it mean that the pressure increasing rate was 0.25 MPa/min?

Author’s Response:

Yes, “a loading rate of 0.25 MPa/min” means that the pressure increasing rate was 0.25

MPa/min. In order to prevent readers from misunderstanding, the loading rate has been re discussed, and the modified content is as follows:

“Force controlled loading is adopted and the pressure increasing rate was 0.25 MPa/min.”